# Are interactions important in estimating flood damage to economic entities? The case of winemaking in France

David Nortes Martínez[1], Frédéric Grelot[1], Pauline Brémond[1], Stefano Farolfi[2], and Juliette Rouchier[3]

[1]G-EAU, Univ Montpellier, AgroParisTech, CIRAD, IRD, INRAE, Montpellier SupAgro, Montpellier, France
[2]CIRAD, UMR G-EAU, 34398 Montpellier, France, Univ Montpellier, 34090 Montpellier, France
[3]LAMSADE, CNRS, PSL (Université Paris-Dauphine), Paris, France

**Correspondence:** Frédéric Grelot (frederic.grelot@inrae.fr)

**Abstract.**

Estimating flood damage, although crucial for assessing flood risk and for designing mitigation policies, continues to face numerous challenges, notably the assessment of indirect damage. It is widely accepted that damage other than direct damage can account for a significant proportion of total damage. Yet due to scarcer data sources and lack of knowledge on links within and between economic activities, indirect impacts have received less attention than direct impacts. Furthermore, attempts to grasp indirect damage through economic models have not gone below regional levels. Even though local communities can be devastated by flood events without this being reflected in regional accounts, few studies have been conducted from a microeconomic perspective at local level. What is more, the standard practices applied at this level of analysis tackle entities but ignore how they may be linked.

This paper addresses these two challenges by building a novel agent-based model of a local agricultural production chain (a french cooperative winemaking system), utilized as a virtual laboratory for the ex ante estimation of flood impacts. We show how overlooking existing interactions between economic entities in production chains can result in either overestimation (double counting) or underestimation (wrong estimation of the consequences for the activity) of flood damage. Our results also reveal that considering interactions requires thorough characterization of their spatial configuration. Based on both the application of our method and the results obtained, we propose balanced recommendations for flood damage estimation at local level.

## 1 Introduction

Floods are natural phenomena that can cause very serious damage, particularly to economic activities (SwissRE, 2017). Due to the impacts of global warming on hydrological regimes and development of territories exposed to flooding, flood damage is indeed expected to increase in the coming decades (Field et al., 2012). It is thus becoming increasingly important to understand the precise mechanisms through which floods cause economic damage. This understanding will help analyze the development of territories exposed to floods, understand the observed – and guess the expected – reactions of agents to the damage they undergo, and improve the design of flood management policies, especially those involving agents' adaptations (Viglione et al., 2014; Grames et al., 2016; Barendrecht et al., 2017; Grames et al., 2017).

It will also be particularly useful to estimate the risk of an exposed territory (risk assessment) and to assess the efficiency of flood management projects, particularly through cost-benefit analyses (Brouwer and van Elk, 2004; Merz et al., 2010; Penning-Rowsell et al., 2013a).

The current way of estimating damage – which may rely on empirical approaches (for example, by examining insurance data), modeling approaches (like damage functions[1]), or a combination of the two – is often limited to assessing direct
damage to buildings and assets (equipment, stocks, furniture). However, only estimating direct damage frequently leads to underestimating the value of the impact (Field et al., 2012). Indeed, as mentioned by many authors (Scawthorn et al., 2006; Meyer et al., 2012, 2013; National Research Council, 1999), impacts other than the direct material ones do occur and should be estimated as indirect damage. Although there is a consensus on the importance to distinguish between direct and indirect damages, what to include in each category is still the subject of debate. For instance, Merz et al.
(2010) consider as direct damage to be the impacts *which occur due to the physical contact of flood water*, and indirect damage to be the consequences of direct damage that occurs *outside the flood event* either in space or time. Meyer et al. (2013) introduced the term *business interruption* for activities that are directly impacted, explicitly restricting direct damage to physical damage, and indirect damage to damage that occurs outside the flood-prone area. Except for the terminology, this is fully compatible with the view of Penning-Rowsell and Green (2000), for whom business perturbation
of directly impacted activities is indirect damage, and other business perturbation is named secondary indirect damage. However, Cochrane (2004) considers business perturbation of directly impacted activities to be direct damage, while indirect damage is any other negative consequences not considered as direct.

Indirect damage has been estimated using either statistic-based approaches (e.g. Kajitani and Tatano, 2014; Yang et al., 2016) or model-based approaches, notably input-output (IO) models (Hallegatte, 2008; Van der Veen et al., 2003;
Hallegatte, 2014; Crawford-Brown et al., 2013; Xie et al., 2012), computable general equilibrium (CGE) models (Xie et al., 2014; Rose and Liao, 2005; OCDE, 2014) or a combination of the two (Donaghy et al., 2007; Rose and Krausmann, 2013; Santos et al., 2014; Hallegatte and Ghil, 2008). Both families have their own strengths and weaknesses. For instance, compared with CGE, IO models are simpler to use and can provide detailed information on economic interdependencies within a regional economy. However they allow neither substitution nor price effects. CGE models are more flexible and
able to account for exogenous interventions in the flood's aftermath and changes in supply/demand. Their drawbacks include the absence of technical limits to substitution – even in the short term – and perfect adjustable markets (Koks et al., 2015; Kelly, 2015; Hallegatte and Przyluski, 2010; Okuyama and Santos, 2014; Przyluski and Hallegatte, 2011). Furthermore, both kinds of models base their calculation of indirect impacts on rather simplistic prefixed coefficients or static ratios of direct damage (Hallegatte et al., 2007; Oosterhaven and Többen, 2017). This method leads to rather large
indirect impacts (Oosterhaven and Többen, 2017) and its accuracy needs improvement (Meyer et al., 2013; Kreibich and Bubeck, 2013).

---

[1]A damage function is a simplified representation of how an asset is damaged by a flood: it draws a link between flood intensity, measured by parameters such as height or duration, and the damage that would be expected if the given asset is flooded by an event of given intensity.

Both kinds of models have been successfully implemented at national and/or regional levels (Bosello and Standardi, 2018; Carrera et al., 2015). However, their potential to provide useful information in decision-making processes when the economic disruptions of floods might vanish before reaching the aforementioned levels is questioned (Green et al., 2011).

There have been attempts to adapt the CGE methodology at local level. For instance, Ferrarese and Mazzoli (2018) used a local social accounting matrix to study the impacts of rural development plans in Mexico. But none have been dedicated to flood damage assessment so far. As flood risk management relies on understanding the consequences of floods (Green et al., 2011) – and flood hazards can be disastrous for local communities and production chains – local flood management practices should benefit from locally-focused studies. Indeed, Meyer et al. (2013) underline the importance

of improving the knowledge of the link between direct and indirect impacts at microeconomic and local levels.

In practice, current methods of estimating flood damage at local and microeconomic level rely on the implicit assumption that economic entities can be treated separately, i.e. without considering how they are interlinked. In concrete terms, damage assessment relies on crossing information on exposure and susceptibility of assets – using geographic information systems (GIS) – that were previously pooled in homogeneous classes (Kreibich and Bubeck, 2013). This practice is

appropriate because it fits the way assets are geolocated by GIS and how damage functions are defined. However, to use these approaches at micro level, without considering the links between economic entities, implicitly assumes not taking the disruption of activities outside the flood-prone area into account.

Moreover, many economic entities are made up of different entities – e.g. different establishments, buildings, plots, etc – located in different places, whose exposure to flooding is not the same. In these cases, defining whether such

systems are directly or indirectly affected by a flood is not easy: some crucial parts may be directly damaged while others remain safe. Taking this internal organisation into account is rare in damage assessment. It is occasionally done in the agricultural sector (Brémond et al., 2013), but has not yet been extended to other economic sectors. In practice, assessing the disruption of business is based on simplistic models or even static ratios of direct damage, whose accuracy needs to be improved (Meyer et al., 2013; Kreibich and Bubeck, 2013).

In this context, we want to introduce the notion of complex productive systems (CPS). A CPS can be an economic entity whose productive components are located in different places, or a collection of economic entities interacting in a global production process (like a supply chain). Among the frequently disregarded flood impacts, in this article, we focus on economic damage due to disturbance of a production process resulting from interactions between different economic entities that may or may not belong to the same firm.

How production processes at local level are affected by flood hazards has not yet been studied in detail. Nevertheless, the literature on business recovery and resilience of economic activities introduces interesting elements (Rose and Krausmann, 2013). *Ex-post* analyses of disasters in supply chains were carried out after the 2011 flood in Thailand (Haraguchi and Lall, 2015; Chongvilaivan, 2012; Linghe and Masato, 2012). Among these analyses, Haraguchi and Lall (2015) showed that damage propagation in a supply chain depends on the location of the productive entities and on the

links between such entities. The same authors also identify the challenges to a better understanding of the robustness of supply chains, namely, the recognition of critical nodes and links, the identification of the direction of links in these com-

plex networks, and the assessment of the effectiveness of bridge ties. This highlights the need for in-depth understanding of the production processes involved and characterization of the links between entities to finely estimate indirect damage at local levels.

Local dynamics are best grasped through *bottom-up* approaches (Crespi et al., 2008): by designing the system from the bottom up, we identify the entities of interest, their interactions and the environment in which they take place. This kind of approach requires specific modeling techniques like agent-based modeling (Tesfatsion, 2002; Smajgl and Barreteau, 2017; Jenkins et al., 2017). An agent-based model (ABM) is a computational tool for the description and dynamic simulation of complex systems. It relies on the description of a system as a collection of autonomous entities, their inter-

actions with one another and their interactions with the environment in which they are embedded (Smajgl and Barreteau, 2014). Additionally, ABMs allow explicit spatial distributions (spatialized models) and time dynamics at different orders of magnitude. ABMs can also be used as complements to other modeling techniques (Jansen et al., 2016) which may help to overcome at least some of the criticisms that IO and CGE models have received. However, even though ABMs are a promising way to improve the estimation of flood impacts (Safarzyńska et al., 2013; Meyer et al., 2013), to date,

applications are rare.

    Within the flood impact research community, it is possible to distinguish already 4 different research trends. The first of these trends would encompass the works of, e.g., Filatova et al. (2009, 2011); Filatova (2015) and Putra et al. (2015). Their main focus rests over the effects of floods on land and housing markets. Specifically land market dynamics (Filatova et al., 2009, 2011) and price formation and urban housing market dynamics (Filatova, 2015; Putra et al., 2015). A second

trend would group the works of, for instance, Haer et al. (2016b,a); Tonn and Guikema (2017) and Erdlenbruch and Bonté (2018). These works focus on household adaptation for flood damage reduction. Pointedly, they focus on the effects on damage estimation of the presence of adaptive human behavior (reduction measures and insurance) (Haer et al., 2016b); the effectiveness of communication strategies and policies to influence households in the adoption of protective measures (Haer et al., 2016a; Erdlenbruch and Bonté, 2018); and flood damage prevention (individually or collectively),

based on risk perception, to evaluate the evolution of flood risk of a city (Tonn and Guikema, 2017). A third trend tackles questions related to insurance in presence of flood risk. Particularly, the implementation of private insurance systems versus government compensations to mitigate financial burdens due to floods in the upper Tizsa river (Dubbelboer et al., 2017; Brouwers and Boman, 2010); or the effects of the UK's flood insurance scheme reform in the London Borough of Camden housing market, its synergies with other flood risk management options and the very sustainability of the

scheme in presence of climate change (Jenkins et al., 2017; Dubbelboer et al., 2017). Last, the fourth trend we will point out, focus on the study of the emergency response to floods, analyzing the effectiveness of incident management practices by evaluating the number of human casualties in case of flood events (Dawson et al., 2011).

    Notwithstanding, in none of the trends identified, works make explicit mention to either agriculture or impact propagation. There exists works regarding the latter nonetheless, as Otto et al. (2017), that propose an ABM to analyze economic

loss propagation along a supply chain for consumers and producers, but focusing on the disruptions of natural disasters in general (not specifically on floods). Also their model is neither spatially explicit nor defined for more detailed resolution

level than regional aggregations. Insofar as the existing ABMs on flood impacts focus on urban areas and direct impacts, a work like this one —focused on a local, agricultural productive chain— is a novelty in nowadays scientific literature.

In this article, we tackle the following question: to what degree can modeling interactions within or between economic entities improve flood damage estimation compared to current approaches that do not take any of these interactions into account? To do so, the article is organized into 6 sections. Section 2 briefly describes the rationale for choosing a cooperative winemaking system (CWS) as a case study of CPS as well as the CWS itself and our data sources. In section 3, we give an overview of our methodology (subsection 3.1), present the model we developed to analyze the impact of flooding on CWS (subsection 3.2), and present the setup and protocols followed in the experiments we conducted (subsection 3.3). The results obtained in our experiments are presented in section 4, with a presentation of the damage estimate for the current practice (subsection 4.1), the influence of introducing interactions (subsection 4.2), and the influence of the configuration of these interactions (subsection 4.3). Section 5 discusses our main results and the main limits of our analysis, and our final conclusions can be found in section 6.

## 2    Case study and data collection

### 2.1    The cooperative winemaking system as a case study of complex productive system

In monetary terms, flood damage to the agricultural sector rarely represents the biggest share of total flood damage. Yet, there is a practical interest in comparing existing ways of estimating damage with those that take into account interactions within the agricultural sector. There are three main reasons for this interest. First, the fact that the damage to agriculture is relatively less important is offset by the fact that agricultural areas may be chosen as targets for floods to protect urban areas (Erdlenbruch et al., 2009; Brémond et al., 2013), meaning that agricultural areas may be negatively impacted. A thorough understanding of how the agricultural sector is damaged is thus crucial when designing compensation schemes due to such risk transfers (Erdlenbruch et al., 2009). Second, the agricultural sector often involves interactions between different economic entities (e.g. farms, suppliers, equipment suppliers, food processing companies, traders). Characterizing the internal organization of these entities is consequently important to accurately estimate how floods affect their activity, even at the level of individual farms (Posthumus et al., 2009; Morris and Brewin, 2014). Finally, Hess and Morris (1988), Morris and Hess (1988) and, more recently, Brémond and Grelot (2012) proposed methods to estimate loss of business by modeling agricultural production systems considering the links between the productive components of a farm (cattle and grassland, agricultural plots and buildings, etc.). This approach, although rare (Brémond et al., 2013) and not yet extended to other economic sectors, deserves further exploration.

Our study is based on two case studies in southern France, where so-called cooperative winemaking systems (CWSs) are very common.

A CWS is a CPS in which two types of economic agents interact: winegrowers (aka farms) and a winery. The cooperative character of the system defines the shared property of the winery's productive means among all winegrowers

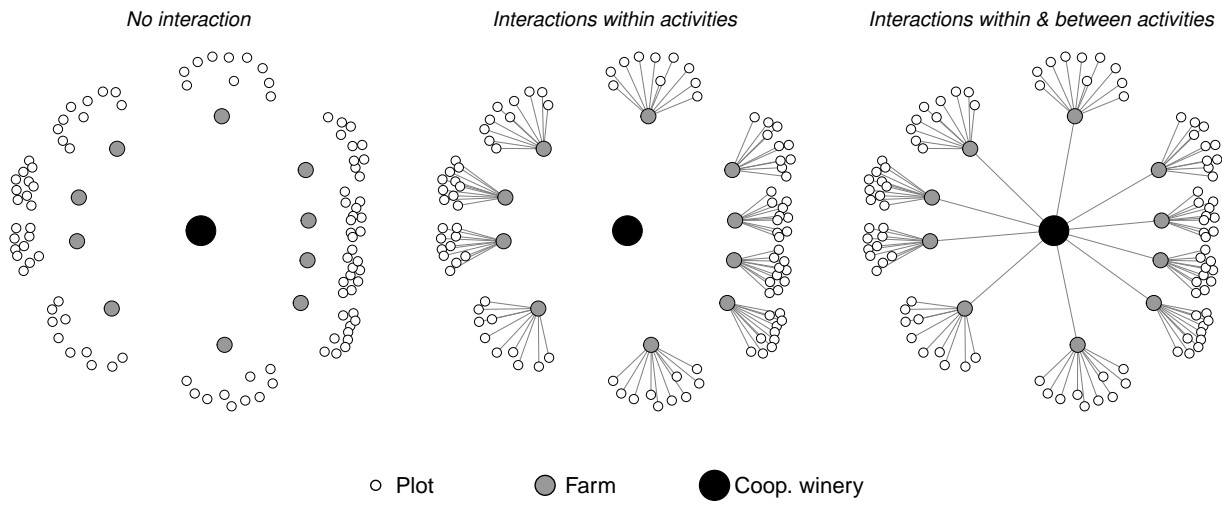

**Figure 1.** Representation of the kinds of interactions and network in the cooperative winemaking system

associated with the winery. Further, all costs, revenues and risks are split among members according to specific rules
drawn up independently by each CWS.

In its most simple version, the system groups a number of winegrowers who, by performing specific tasks in their
vineyards (aka plots) all year round, harvest a specific yield of wine grapes every year. The yield is then transported
to the winery, where the system's wine production is centralized and stored. The winery also sells the stored wine and
distributes the yearly profits among its associated winegrowers.

In this simple formulation two kinds of interactions can be distinguished: *interactions within activities* and *interactions*
*between activities*. The former represent the inter-dependency of the different components of the farm (vineyards and
buildings) that are spatially dispersed, whereas the latter represent the inter-dependency between the winegrowers and
the winery. Despite their names, both cases should be understood as flows of information from one entity (plot, farm,
winery) to another. Namely, when an explicit link between two material entities exists, the entities dispose of information
about their own state and each other's state. These interactions plus the shape of the CWS's network are illustrated in
figure 1.

### 2.2 Data collection

We collected data from the Aude and Var administrative departments (southern France), both subject to major floods
that have impacted the winegrowing sector (Vinet, 2003; Bauduceau, 2001; Collombat, 2012; Chambre d'agriculture Var,
2014). Data were collected from several sources in order to identify common patterns and plausible hypotheses related
to CWSs.

The sources include qualitative interviews with winegrowers and heads of cooperative wineries (RETINA, 2014-2016) in both departments, that provided useful insights into soil productivity, the stages of production, the behavior of the agents, plausible business sizes and governing rules. On these topics we also counted on the works of Biarnès and Touzard (2003); Chevet (2004); Agreste (2010); Battagliani et al. (2009) and FranceAgriMer (2012). The sequence of technical operations carried out by winegrowers is based on technical information provided by the Chamber of Agriculture.

Financial data and data related to price, costs and cost structures came from Folwell and Castaldi (2004); Centre d'économie rurale (2014, 2017); Chevet (2004); FADN (2014); INSEE (2016); CCMSA (2017); Brémond (2011) and FranceAgriMer (2012). Flood material damage was modeled using existing damage functions adjusted to the local context from Brémond (2011) and Rouchon et al. (2018). Lastly, patterns of exposure where obtained using geographical information from IGN (2020) and MTES (2020)

Table B1 in appendix B further discloses these references and the main area(s) in which the information obtained has been relevant.

## 3  Method

### 3.1  Overview

The approach we follow in this paper is outlined in figure 2. To determine whether or not to take into account the existing interactions between material entities influences the estimation of flood impacts, our work uses a comparative method. We compare the simulations obtained from two experiments (subsection 3.3) carried out within the virtual laboratory provided by the COOPER model (Nortes Martinez et al., 2019).

The COOPER model (Nortes Martinez et al., 2019) is an agent-based model (ABM) we built to serve as a virtual laboratory for the ex-ante estimation of impacts of a wide variety of flood phenomena over a CWS (subsection 3.2)[2]. With this model we seek to mimic from the bottom up the functioning a specific system (the CWS), composed by entities of different nature (see next section) that interact with each other in a specific, organized way to obtain a final product. This way, the output of the system depends on the performance of each individual and on the interaction between them all. By observing the state and the performance of each individual entity at any moment, the COOPER model enables us to evaluate the impacts of floods and to track down the origin, the persistence and the length of disturbances within the CWS. Namely, we are able to track down the origin and extent in time, space and aggregation of individuals of a disturbance within the bounds of the CWS without assigning ratios or weights to direct impacts to calculate indirect impacts (far more common method in economic modeling).

The absence of predetermined impacts (direct impacts) upon which apply weights to calculate indirect impacts, additionally enables us to accommodate the different impact classifications existing in the literature. In our case, we choose to adscribe to a classification that is going to consider direct impact everything that is directly exposed to the flood except

---

[2]Further additional details are provided in the model documentation, available online at the CoMSES computational model library (Rollins et al., 2014).

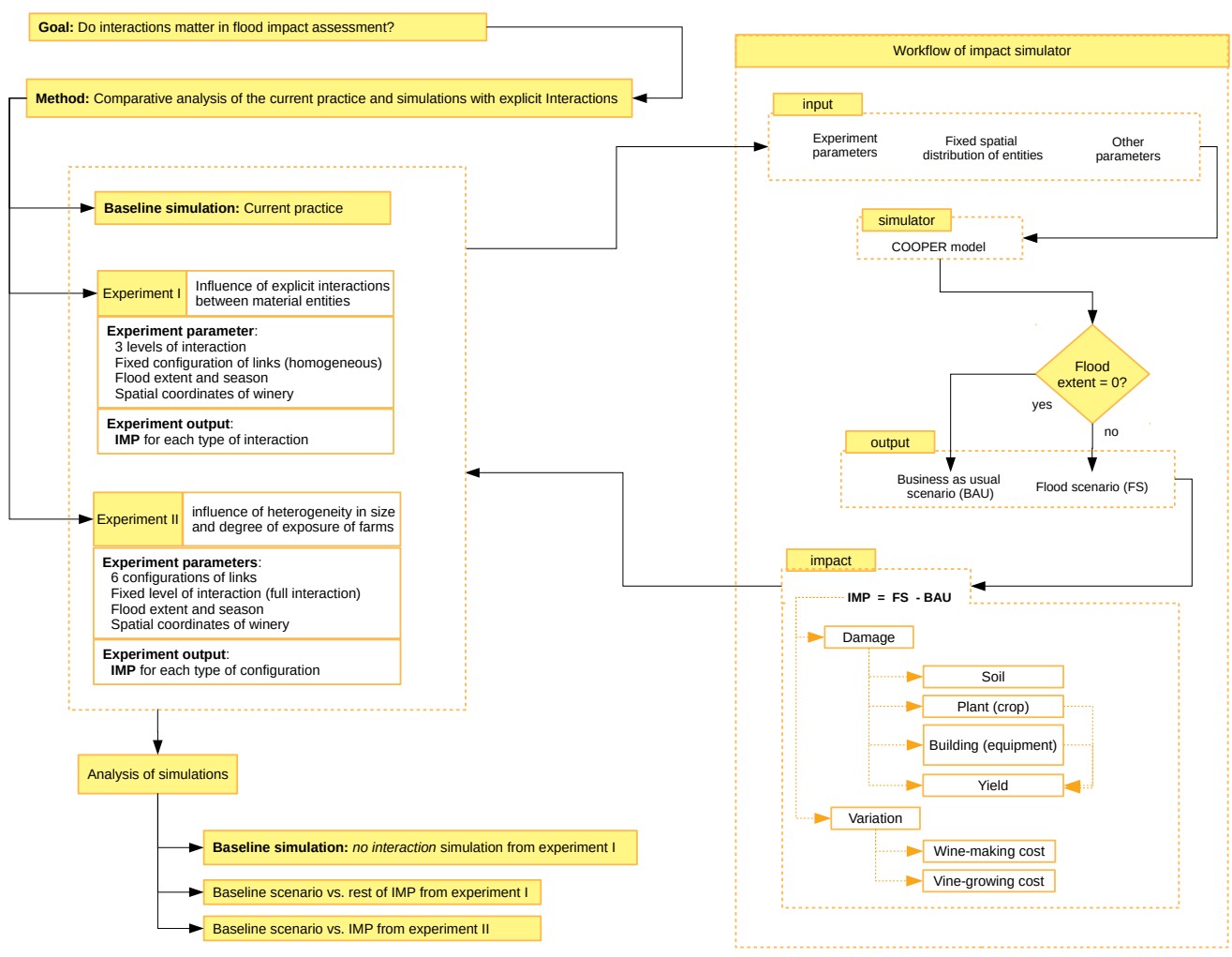

**Figure 2.** Workflow of the approach used in the study, detailing the study's goal, method, number of experiments and impact simulator's workflow. For each experiment, goal(s), parameters and output have been explicitly included

for the loss of yield. This loss and all the effects caused by the direct exposure to the floods are considered as indirect impacts. The explanation rests on the fact that the farms' yield is considered an intermediary product within the CWS. To

210 calculate the added value lost, the yield should, first, be transformed in the winery, then sold in the market. This lack of immediacy in the materialization of the impact motivates our classification as indirect impact.

Our choice of using ABMs to build the COOPER model is based on several reasons. First, ABMs allow us to take into account specific topologies to link entities to one another (which in turn define their interactions). Also, ABMs allow us to

test whether the influence of the concrete geographic locations of agents have some repercussion when estimating flood impacts. Third, the explicit introduction of the time scale enables us to observe how the interactions between different entities affect them in different terms. Last, ABMs allow us to test the system, getting information on its responses to disturbances and the mechanisms that guide them. Hence, ABMs display great potential to improve our knowledge on how impacts are triggered and how they spread out in economic systems and productive chains from a micro base.

In the virtual laboratory provided by the COOPER model we conduct two different experiments. The first experiment (subsection 3.3.1) compares the estimate of flood impacts carried out following the usual practice (*no interaction*) with impact estimates made taking into account different degrees of interaction (the so-called *partial* and *full*). All these simulations are run on a fixed spatial distribution of material entities homogeneously linked to each other, making it possible to ensure that the source of variation is the presence of interactions. Furthermore, the experiment is conducted in events of varying magnitude, so potential non-linearity can be evaluated.

The second experiment (subsection 3.3.2) builds upon the previous one, and compares the usual practice with estimates of impacts generated by linking the material entities in a heterogeneous way. In this experiment, only the greater degree of interaction (*full*) is considered.

The two experiments are carried out using two alternative locations of the winery building. The objective is to show the effect of flooding or not of this central element of the system.

With the results obtained, we set up an index of differences. To build it, we start from the absolute damage obtained following the usual practice (subsection 4.1). Using this simulation as a baseline, we calculate the percentage that represents the difference in the series of estimated damages using the alternatives proposed in experiments I (section 4.2) and II (section 4.3).

## 3.2 Model

The COOPER model is built upon the description of the system provided in subsection 2.1. We modeled three types of material entities: farm land plots, farm buildings and the winery's building. These material entities are located in a virtual territory. At the same time, this territory is divided into cells that can host one and only one material entity.

Upon these three material entities we identify two kinds of agents: farms and winery. Each farm (understood as an economic agent) is formed by the combination of a number of plots and a farm building in which all the farm equipment, stocks and harvested products are located. The winery (understood as an economic agent) is represented by only one building in which all equipment, stocks, products are assumed to be located. The material components of the winery are assumed to be one indistinct material component in one location, consequently any interactions *within activities* for the winery are not taken into account.

In the COOPER model each time step represents one season. For both farm and winery, we count on simplified —and seasonally adjusted— versions of their own real-life complex schedules linked to biological cycles of vines. As a result, the global internal schedule in the model is given by the coexistence and interaction of those individual schedules. To

illustrate the point, figure 3 outlines the global model schedule and each agent's own schedule. A year begins in winter and ends in autumn.

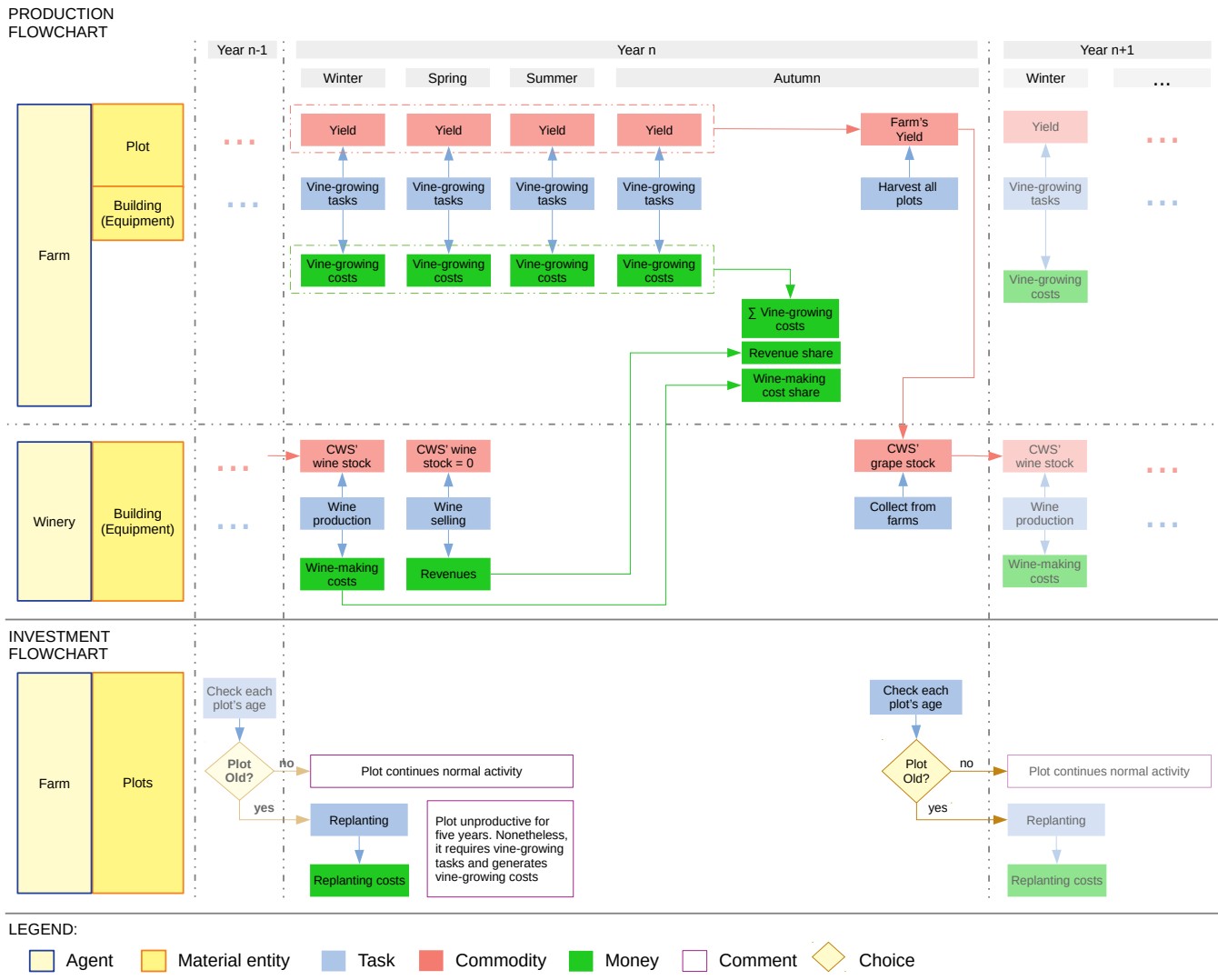

**Figure 3.** Overview of production and investment seasonal schedules in the COOPER model. Both schedules disclose tasks as well as commodities and monetary flows per season and year (also choices in the case of investments)

The production dynamic goes as follows: Farms perform tasks upon their plots which, on the one hand, enable each farm to obtain a yearly yield, and, on the other hand, provoke the apparition of vine-growing costs. These tasks take place during the four seasons. Each autumn, farms harvest their plots and move their yields to the cooperative winery. During the next winter, the cooperative produces wine with the yield obtained from the farms and commercializes the production

in spring. Once all the wine is sold (in spring), the cooperative winery splits both revenue and wine-making cost among farms proportionally to the yield they provided in the prior campaign (see appendix A1). It is worth noting that there is a time gap between the transfer of the yields from the farmers to the winery and the transfer of revenues from the winery to the farmers. Therefore, there exists a time gap between the revenues the farmer is perceiving and the costs that they are effectively financing.

As it happens in real CWS, plots present heterogeneous ages. In the COOPER model, plots reaching 30 years of age get replanted. The replanting takes place at the end of autumn and replanted plots remain unproductive for 5 years (20 time steps). Apart from the clear cost in terms of yield harvested during the their first five years of life, replanting itself also bears monetary costs for farmers.

### 3.2.1 Flood process: intensity and impacts of floods

The COOPER model does not pretend to recreate a specific event. Hence, estimated impacts are not based on recorded or observed costs but on synthetic damage functions. Notwithstanding, the COOPER model relies on real data to establish the way the system works and plausible losses for individual entities (e.g. losses of yield in plots). In other words, due to the limitations in data availability at our resolution level, the COOPER model mixes both *data-driven* and *expert-driven* approaches to build a geolocated, computational laboratory for exploratory research. Therefore, it enables us to simulate the impacts of a large variety of flood events over the same system.

The virtual territory in which material entities are located is divided into two different areas: one subject to floods (flood-prone area), one not. In the COOPER model, floods are defined by two parameters: extent and season of occurrence. Flood extent is measured along the $X$ axis in the interval $[0, 100]$ assuming the river is located in $X = 0 \; \forall \; Y$. So, for instance, a spring flood of extent 50 impacts all cells located in the band [1–50] in the flood-prone area in spring. In the study we are presenting here, only one flood can occur over the whole simulation period, thus a spring flood designates a flood occurring in the first spring after the beginning of the simulation.

When a flood hits the system, it causes direct material damage to the material entities –farm plots, farm buildings and winery building– which may also disrupt the productive process in different ways, affecting the economic agents –farms and winery. Appendix A3 includes a mathematical formalisation and hierarchization of the processes and damages we present below.

At plot level, material damage is threefold: i) damage to the soil, considered independent of the season; ii) damage to yields, dependent on the season; and iii) damage to vines, stochastic and dependent on the season (destruction of the vines depends on a probability function, which is not the same all year long). We consider necessary to distinguish winter from the ensemble of spring-summer-autumn and, within this ensemble, the case where vines are not destroyed from the case when vines are destroyed.

Winter is a special case. When plots are hit directly, the sole impacts that floods produce is soil-reconditioning. It provokes a direct financial impact over farms who own impacted plots (benefits will decrease as a consequence of the

extra reconditioning cost), but not further damages over yield nor vines, therefore no reduction on production and on revenues will take place.

Concerning spring-summer-autumn, in the case that floods do not destroy the vines in the flooded plots, the harvest is lost in a variable amount linked to the season. Also, soil-reconditioning tasks should be performed. At farm level, the yield harvested will depend on the number of plots flooded. At the same time, plots whose yield is completely lost, save vine-growing cost to the farm. At winery level, as it happens at farm level, the yield collected will be affected by the number of plots hit owned by the winery's associates, and so will be the annual production and the sales. Ultimately the financial balances of the winery and the farms will reflect the impacts of the flood.

If vines are destroyed when plots are flooded, the consequences have further ramifications: at plot level, both the vines and the whole harvest is destroyed. At farm level, as in the prior case, yield lost depends on the number of plots flooded and vine-growing cost are saved. Vine destruction also introduces a longer term effect: destroyed plots need to be replanted. Assuming they are replanted immediately (next winter), they will need 5 complete years to be considered productive. Therefore, *ceteris paribus*, the farm does not only loses the harvest of the current campaign but the harvest of the next 5 years per plot destroyed. At winery level, those longer term impacts will be reflected too.

Vine growing cost savings appear because, whether it is due to the destruction of the vines or to direct damage to the yield, as soon as the plot loses all its yield, the farm stops performing winegrowing tasks during the current campaign in the plot concerned thereby saving the cost of the remaining tasks until the beginning of the following campaign.

As for time spans, damages in soils, harvest and variations in vine-growing costs are accounted to $t = 1$. If vines are not destroyed, variations in production, ergo in revenues and wine-making costs, will be accounted to $t = 2$ (thus delayed one year); if vines are destroyed, impacts on production, revenues and wine-making costs will last until $t = 7$ assuming plots are replanted in $t = 2$ (otherwise impacts will last longer).

In addition to impacts on plots, farms can experience impacts that can be split into two kinds of consequences: consequences due to buildings and materials flooded, and, once it happens, consequences due to the coping strategy chosen. Farms are assumed to be motivated to preserve their *status quo ante*. This means that, in absence of constraints, buildings will be repaired and materials replaced right away, so the farm is fully operational next season[3]. The same principle applies to plot's replant: in the absence of constraints, it is done the first winter season following the flood. But when the building is hit, we assume that part of the vine-growing material is lost/hit. Farms, consequently, will have to pay for reparations and, additionally, they cannot fully perform their seasonal tasks. To cope with this situation, they can choose between two tactics:

The first one, hereafter labeled *external* tactic, states that farms whose buildings are hit by a flood can hire external service providers to perform the task in its place. Such strategy saves all the yield in plots since the tasks are fully performed, but increases the seasonal vine-growing costs. The alternative tactic, henceforth referred as the *internal* tactic, establishes that the farm counts on its own resources to perform the seasonal tasks. Since part of the material is

---

[3]After the flood hits the farm in the beginning of the season, we assume that, in the absence of financial constraints, farms have enough time during the season to repair and be fully operational next one.

lost, we assume the farm can only perform the half of the tasks planned for the season. As a consequence, seasonal
vine-growing cost decreases by 50% but there is an associated lost in yield.

The time span for impacts derived from the choice of tactics is different: assuming the flood hits the system in year
$t = 1$, effects over vine-growing costs become part of impacts in $t = 1$, while effects over yield derived from the *internal*
tactic will be felt in year $t = 2$, once the yield is processed and the wine produced and sold. Both tactics eventually affect
financial balances but, while the *external* tactic limits impacts to the year in which flood hits the system, the *internal* tactic
generates more persistent impacts.

As it happens for farms, impacts over wineries have a twofold nature: first, regardless of the season, when a cooperative
winery is hit by a flood, the model assumes damages in buildings and equipment. Second, these damages also affect
the capacity of the winery to perform its normal activities. Concretely, when the winery gets hit during winter, the material
damage suffered impedes the processing of the yield collected during the prior campaign, thus the wine production. With
330 nothing to sell [4], there are no revenues for farmers and the wine-making cost is reduced to the winery's fixed costs[5].

Insofar all production and sales are done in and through the cooperative winery, all the associated farms will lose
all production and revenues. They will be imputed, though, with their share of the winery's fixed cost and reparations.
Financial balances will reflect such situation.

If the winery is flooded in spring, we consider the wine-making processes already finished and the production ready
to be sold. However, material damages will make the winery lose the production and, as in winter, no revenues over the
yield of the prior campaign will be perceived. Contrary to winter, in spring, since wine-making activities are done, farms
will be imputed with all the wine-making cost corresponding to its share plus the reparations needed.

During summer season, wineries are not expected to perform any essential task. Therefore, when they are flooded,
impacts are limited to reparations, with no further effect besides the ones over the financial balance of the winery and its
associated farms.

Floods over the winery's buildings in autumn, hinders the winery from collecting the yield coming from its associated
farms. Under such circumstances, all farms lose their yields. This fact prevents the system from having input to produce
wine during winter of the following campaign. Without production, effects are the same as the already described situation
for winter, but delayed by one period: no sales, ergo no revenues, and wine-making cost reduced to the fixed cost.
Concerning the imputation of costs from the winery to its associated farmers, we can identify two different mechanisms:
the first one is when the winery is flooded, but production can be done or has been done. In such case, reparation costs
are imputed among associated farms proportionally to the yield provided by each farm (see appendix A2). In such regard,
it is worth noting that, inasmuch as fixed costs exist in the structure of costs of the winery, cost-revenue sharing rules
create an implicit interaction among all the farms in the CWS: if one farm loses its whole harvest, it will not receive any

---

[4]Since floods happen at the beginning of the season, the winery will have time to be fully functional for the next season, and to perform sales.
However, to not be able to produce the wine, has left it with no production to be sold.

[5]Winemaking costs are twofold: fixed and variable.

revenue from the winery, but neither will it have to pay its "normal" share of fixed costs. All other farms will consequently be indirectly impacted because they will now have to pay that share of fixed costs.

The second mechanism is triggered only when the production-commercialization process gets disrupted, and production cannot be done. In this case, wine-making cost is reduced to the winery's fixed cost. Added to reparation costs, both are imputed proportionally to the number of farmers (see appendix A2).

### 3.2.2 Flood impact calculation

In the COOPER model, the CWS rests, both at collective and individual levels, over a vector of four key variables: *investments and reinvestments —$I_t$—, production —$Q_t$—, vine-growing costs —$C_{vg}$—* and *wine-making costs —$C_{wm}$.* The variable $I_t$ serves us to group all reparations to be done in the system after a flood, reinvestments in vines and materials and, also, planned investments independent of the flood.

Every time a material entity in the CWS is flooded, one or more of those variables are going to experience a change. Thus, assuming that $\boldsymbol{BAU}_t$ and $\boldsymbol{FS}_t$ are two vectors of key variables for their respective business as usual scenario (BAU) and flooding scenario (FS):

$$\boldsymbol{BAU}_t = (I_t, Q_t, C_{vg_t}, C_{wm_t}) \tag{1}$$
$$\boldsymbol{FS}_t = (I'_t, Q'_t, C'_{vg_t}, C'_{wm_t}) \tag{2}$$

We can define the impact of a flood at any moment $t$ as (see appendix A4 for more details):

$$\boldsymbol{Imp_t} = \boldsymbol{FS}_t - \boldsymbol{BAU}_t \tag{3}$$

In the COOPER model, information on those four key variables is recovered through a collection of 10 different indicators (see table 1). These indicators are available for every individual farm in the CWS at any time step. Insofar as the collectivity –the CWS– in the COOPER model is the result of the aggregation of the individuals –the farms– rather than an extrapolation, the same 10 indicators (thus the four key variables) available at the farm level are also available for the CWS as a whole by means of the aggregation of the individual values.

In table 1 we also display the classification of impacts that we are using in this work. As we can see, all impacts labeled as direct (thus coming from direct exposure to the flood) also belong to the key variable *investments and reinvestments* ($I_t$). On the contrary, indicators belonging to key variables *production* ($Q_t$), *vine-growing costs* ($C_{vg}$) and *wine-making costs* ($C_{wm}$), inasmuch as the production process should be finished for them to materialized, are considered indirect impacts within the boundaries of the CWS.

| Indicator | Metric | Key variable | Agent | Classification |
|---|---|---|---|---|
| Damages in soils | Cost of soil reconditioning | $I_t$ | Farm | Direct |
| Damages in plants | Cost of replanting corrected by the modification of the reinvesting schedule** | $I_t$ | Farm | Direct |
| Damages in harvest due to floods | Market value of wine that would have been produced | $Q_t$ | Farm | Indirect |
| Damages in harvest due to plant destruction | Market value of wine that would have been produced | $Q_t$ | Farm | Indirect |
| Variations in vine-growing cost | Variations of cost due to variations of yield and choice of coping tactic | $C_{vg_t}$ | Farm | Indirect |
| Damages in harvest due to damages in farm | Market value of wine that would have been produced | $Q_t$ | Farm | Indirect |
| Damages in farm's equipment | Reposition cost | $I_t$ | Farm | Direct |
| Variations in wine-making cost | Variations of cost due to variations in yield from farms | $C_{vm_t}$ | Winery | Indirect |
| Damages in winery's equipment | reposition cost | $I_t$ | Winery | Direct |
| Damages in harvest due to damages in winery | Market value of wine that would have been produced | $Q_t$ | Winery | Indirect |

**This correction takes into account the difference between destroying a newly replanted vineyard or fully amortized vineyard that was going to be replanted anyway

*Key:* $I_t$ = Investment | $Q_t$ = Production | $C_{vg}$ = Vine-growing cost | $C_{wm}$ = Wine-making cost.

*Remark:* Our indicators consider discount factors to assess damages along time. Market values are calculated at a constant price

**Table 1.** Indicators of impacts of floods in the COOPER model. The table includes the indicators implemented in the COOPER model to measure the impacts of floods, their metrics, the key variable under which they are grouped and the main agent in which they find their origin. As well, each indicator is classified as direct or indirect, according to the classification of section 3.1. All indicators are measured in euros (€)

### 3.3 Simulation protocol and experiments

To answer to our question, we performed a twofold experiment designed to analyze the divergence of estimated flood impacts, simulated with the COOPER model, according to different schemes of interactions between economic entities.

Regardless of the interaction scheme used for each particular experiment, all simulations share the same configuration to avoid sources of unwanted variation. This general setup presents the following characteristics across simulations.

The CWS used for simulations in our two experiments is composed of 51 agents and 551 material entities: One *winery* agent with a winery building; 50 *farm* agents each with a farm building,, and sharing a total of 500 plots of size 1 ha. The CWS thus mobilizes 500 ha of productive land.

Farm buildings and plots remain in the same location for each simulation. This ensures that the physical components of the farms are always impacted in the same way by a flood of a given extent and at a given season (see Table 2). Specifically, 20% of the farm buildings (i.e. 10 out of 50) are located in the flood-prone area, randomly distributed within the band [30–100] (see Figure 4), whereas 30% of the plots (i.e. 150 out of 500) are in the flood-prone area, randomly distributed within the band [10–100] (see Figure 4).

Concerning the location of the winery building, we performed two sets of analysis: one with the winery building located at position 1 in the flood prone area (and consequently always flooded), one with the winery building located outside the flood prone area (and consequently never flooded).

Productive activities in the CWS are simulated for time spans of 30 years, divided in 4 seasons.

#### 3.3.1 Experiment I: Influence of the presence of explicit links between material entities

Our first experiment targets methods of flood damage estimation and the influence that they have on the resulting impact. Specifically, we search to estimate the influence that the inclusion of explicit interactions between material entities has on the estimation of flood damage. To do so, we sequentially simulate and compare three alternative cases (figure 1). First, the *no interaction* case, which corresponds to current practices in damage assessment, and includes no explicit links between material entities, thus no explicit interaction exists and each entity only has access to its own state. Next, the

*partial interaction* case, in which only *interactions within activities* are explicitly included. Namely, farms and plots have access to both their own states and each other's states, whereas the winery only has access to its own state. Last, the

**Table 2.** Common characteristics for the location of material components in the simulations

| Element | Number of elements in | | Total |
| --- | --- | --- | --- |
| | Flood area | Safe area | |
| Winery | 0 or 1 | 0 or 1 | **1** |
| Farm | 10 | 40 | **50** |
| Plot | 150 | 350 | **500** |

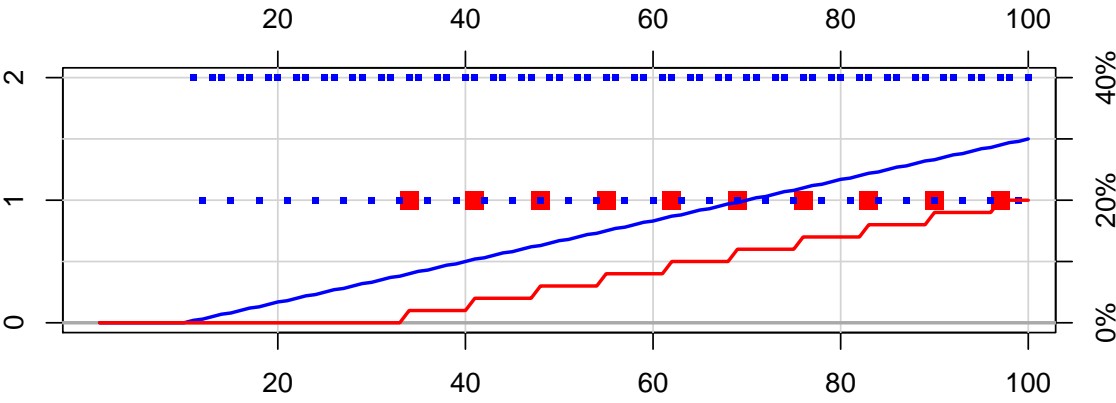

*Legend*: blue represents plots, red farm buildings. The x-axis indicates the extent of the flood. The left y-axis gives the number of elements located at the position of the corresponding points. The right y-axis shows the corresponding cumulative percentage given by the lines. For example, 10% of the plots (blue lines) are located in the floodplain at a position of 40 or under, 2 plots are present at this precise position.

**Figure 4.** Spatial distribution of farms buildings and plots inside the flood-prone area

*full interaction* case, which accounts for interactions both *within* and *between* activities. Accordingly, all material entities in the system have access to their own states and each other's states.

In both *no interaction* and *partial interaction* cases, due to the lack of explicit links, information on other entities' states does not flow throughout the system. In these two cases, additional assumptions are required to perform damage estimation and business disruptions.

The first one, referred as **A1**, is an implicit assumption in current practices of damage assessment and concerns only the *no interaction* case. It can be stated as "*Independently of whether farm buildings have been flooded, winegrowing tasks can all be accomplished in non-flooded plots at the normal cost, as if the buildings of the corresponding farms had not been flooded*". In other words, under assumption **A1**, grape production depends only on what happens in the plots (damage to plots and damage to the farm buildings is estimated separately at farm level).

The second and third assumptions concern both the *no interaction* and *partial interaction* cases and are still part of current practices of flood damage assessment. The first of them, referred as **A2**, states that *the winery receives the quantity of grapes computed as if no farm buildings or plots were flooded*, whereas the second one (referred as **A3**) specifies that "*The cost of wine production is computed as if no farm buildings or plots were flooded.*" Under these two last assumptions, wine production and sales depend only on what happens to the winery building, while damage to plots and farms buildings can be estimated separately at farm level. Table 3 sums up which assumption(s) applies to each case.

A complete list of initialization values and endowments is available in the COOPER description in the CoMSES computational model library.

In contrast, in the *full interaction* case, where all links are explicit, there is no need for those assumptions. Regarding assumption **A1**, in the *full interaction* case, whether the tasks required in the plots are performed or not depends on the state of the farm building (flooded / not flooded) and the coping tactic used by the farm concerned. Regarding assumptions **A2** and **A3**, in the *full interaction* case, the quantity of grapes the winery receives depends on the effective damage to the farm, which also makes it possible to calculate the loss of wine products and impacts on winemaking costs.

For this experiment, we complete the general setup mentioned above with the following configuration: all the farms are the same size (10 plots) and the same proportion of plots are located in the flood-prone area (around 30%). This configuration is labeled *homogeneous* configuration (See table 4 for more information).

The results of experiment I are presented in section 4.2 as a comparison of the estimated impacts using the *no interaction* case (standard practice) as baseline.

### 3.3.2 Experiment II: Influence of agent heterogeneity in flood damage estimation

This experiment is designed to test whether heterogeneity in farm size and degree of exposure of farms has an impact on the amount of damage suffered by the system in the case of flooding.

To introduce these two factors of heterogeneity without modifying the spatial distribution of material entities, we construct different configurations of links between plots and farms' builings by modifying which plots belongs to which farm. These configurations are as follows (see figure 5 for a schematic representation and table 4 for the main characteristics of the spatial distribution of the different configurations):

**homogeneous**  (figure 5a) All farms have the same number of plots (10). The proportion of plots in the flood-prone area is the same in all farms.

**size**  (figure 5b) 10 farms are big (30 plots), 40 farms are small (5 plots). The proportion of plots in the flood-prone area is equivalent for each farm.

**exposure-worst**  (figure 5c) All the farms have the same number of plots. The farms whose building is located in the flood-prone area have all their plots located outside the flood-prone area. The plots in the flood-prone area are approximately equally distributed among the remaining farms.

**Table 3.** Modalities of interactions and assumptions for damage estimation

| Case | Assumptions |
| --- | --- |
| *No interaction* | A1 + A2 + A3 |
| *Partial interaction* | A2 + A3 |
| *Full interaction* | — |

***exposure-best*** (figure 5d) All the farms have the same number of plots. The farms whose building is located in the flood-prone area also have all their plots located in the flood-prone area. The remaining plots in the flood-prone area are approximately equally distributed among the remaining farms.

***size-exposure-worst*** (figure 5e) 10 farms are big (30 plots), 40 farms are small (5 plots). All big farms' buildings are located in the flood-prone area whereas all their plots are located outside the flood prone area. All the plots in the
450 flood-prone area thus belong to small farms, whose buildings are located outside the flood-prone area.

***size-exposure-best*** (figure 5f) 10 farms are big (30 plots), 40 farms are small (5 plots). The building and all the plots belonging to 10 of the small farms are located inside in the flood-prone area. The remaining plots located in the flood-prone area belong to the remaining small farms. The plots and buildings of the 10 big farms are located outside the flood-prone area.

We simulate the damage in the *full interaction case* for each of these configurations and compare them to the same baseline as in the experiment I (*no interaction* and *homogeneous*[6]). The results are analyzed in section 4.3.

---

[6]When damage are assessed in the *no interaction* case, there is no influence of the configuration of links, because of assumption **A1**.

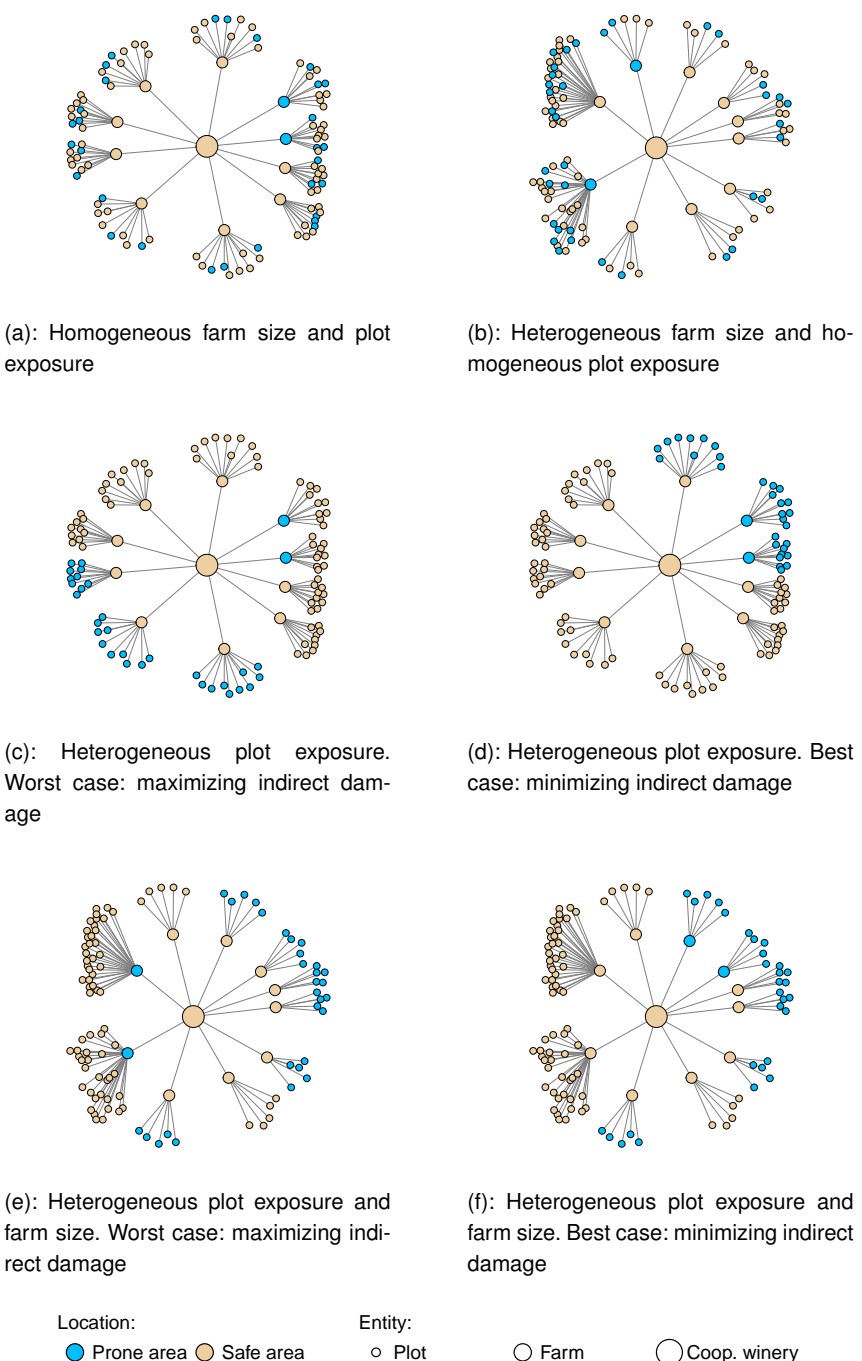

(a): Homogeneous farm size and plot exposure

(b): Heterogeneous farm size and homogeneous plot exposure

(c): Heterogeneous plot exposure. Worst case: maximizing indirect damage

(d): Heterogeneous plot exposure. Best case: minimizing indirect damage

(e): Heterogeneous plot exposure and farm size. Worst case: maximizing indirect damage

(f): Heterogeneous plot exposure and farm size. Best case: minimizing indirect damage

Location: ● Prone area ● Safe area    Entity: ○ Plot ◯ Farm ◯ Coop. winery

**Figure 5.** Schematic representations of the configurations of links allowing us to introduce two sources of agent heterogeneity: farm size and flood exposure

**Table 4.** Comparison of the main spatial distribution characteristics of the different configurations of links

| configuration | size | exposure | $n_{farms}$ | building | $n_{plots}$ | exposed plots |
|---|---|---|---|---|---|---|
| *homogeneous* | homogeneous | homogeneous | 10 | exposed | 10 | 32% |
| | | | 40 | safe | 10 | 30% |
| *exposure-best* | homogeneous | heterogeneous | 10 | exposed | 10 | 100% |
| | | | 40 | safe | 10 | 12% |
| *exposure-worst* | homogeneous | heterogeneous | 10 | exposed | 10 | 0% |
| | | | 40 | safe | 10 | 38% |
| *size* | heterogeneous | homogeneous | 8 | exposed | 5 | 38% |
| | | | 32 | safe | 5 | 32% |
| | | | 2 | exposed | 30 | 33% |
| | | | 8 | safe | 30 | 28% |
| *size-exposure-worst* | heterogeneous | heterogeneous | 0 | exposed | 5 | – |
| | | | 40 | safe | 5 | 76% |
| | | | 10 | exposed | 30 | – |
| | | | 0 | safe | 30 | – |
| *size-exposure-best* | heterogeneous | heterogeneous | 10 | exposed | 5 | 100% |
| | | | 30 | safe | 5 | 66% |
| | | | 0 | exposed | 30 | – |
| | | | 10 | safe | 30 | 0% |

*Remark*: the first column gives the name of the configuration, the column "size" (respectively "exposure") indicates whether this configuration is considered as homogeneous or heterogeneous in terms of size of farms (respectively in terms of proportion of plots exposed to flood). The following columns give quantitative information. For the corresponding configuration, there is $n_{farms}$ farms that have their building in the situation given by the column "building", each connected to $n_{plot}$ plots. The proportion of exposed plots belonging to these farms is given in the column "exposed plots".

### 3.3.3 Simulations performed

For both experiments, simulations are run for flood extents from 15 to 100 – increasing at a step of 5 – for each combination of farms' coping tactic, and season. As the COOPER model includes stochastic processes, each flood scenario is replicated 50 times.

For experiment I, the combinations are completed with each case for the location of the winery and the type of interaction. For experiment II, the combinations are completed with each case of configuration of links.

These experimental designs result in a total of 43,200 different simulations for experiment I and 86,400 different simulations for experiment II.

## 4  Results

### 4.1  Baseline

Figure 6 shows the absolute flood damage for the CWS according to the current practice. The extent of the damage depends mainly on whether the winery was flooded or not, due to the importance of the equipment in the cellar. The damage also differs greatly depending on the season of the flooding. The damage increases in proportion to the flood 470 extent, which reflects the increase in the number of flooded material entities.

Table 5 displays the minimum and maximum amounts of damage endured by the CWS in the baseline scenario, disclosing direct damages from total damages. Values are displayed by season and winery position. Also, in order to facilitate the interpretation of the magnitude of the estimated damage in relation to the CWS, table 5 presents a relative measure of damage taking as reference the annual potential added value of the system (PAV hereafter). This measure is 475 defined as the added value that would be obtained in the CWS if all plots in the system were productive in the same year. The PAV value for our CWS is € 794,000. Thus, when we say that, e.g., the largest-scale spring floods cause a relative total damage of 2.258 times the PAV in the case that the winery is safe, it means that the CWS would need the production generated in 2.2 years working at plain capacity to cover the damages.

As we can see, damages are orders of magnitude bigger when the cooperative winery is flooded, accounting for up 480 to 10 times the PAV. When the winery is not flooded damages are not only smaller. There is also a reorganization of the impact by season: when the winery is not flooded, the lowest impact by flood event arrives in winter instead in summer. The amount of total damages that correspond to direct impacts in our baseline vary depending on seasons, whether the winery is flooded and the extent of the flood event.

It is worth noticing at this point that, due to the consistency regarding system size and spatial distribution of material 485 entities that we use across simulations and experiments, the magnitude of the direct impacts in the CWS is going to be constant throughout simulations and experiments. Thus the variations in the estimated impacts of floods presented in the following sections originate in the indirect impacts.

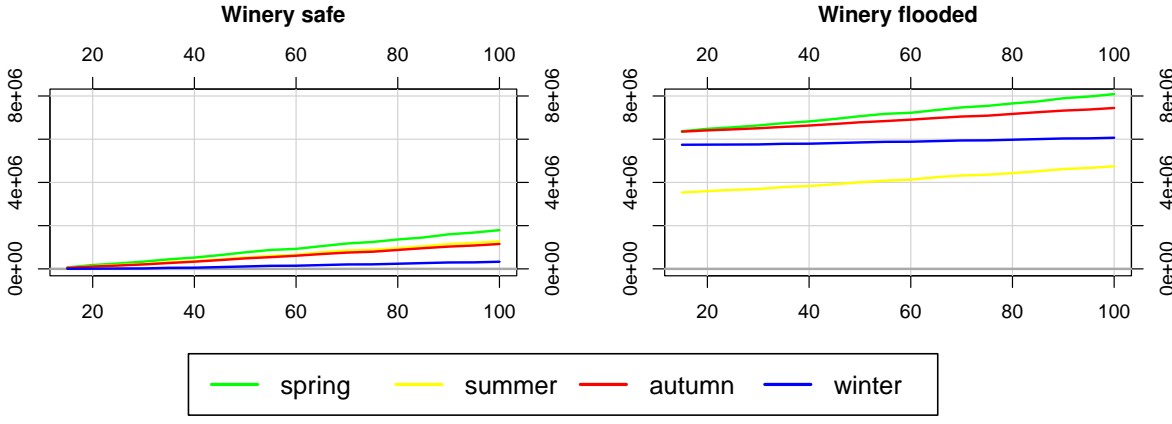

Remark: In each figure, the x-axis indicates the extent of the flood; the y-axis corresponds to the absolute damage in euros.

**Figure 6.** Absolute damage for the baseline simulation

## 4.2 Influence of interactions on damage estimation

In this section, we analyze the importance of accounting for interactions between the entities of a CWS in estimating
flood damage. The results are shown in Figure 7. In the figure, the different lines show the relative difference in damage between cases of *partial interaction* (dashed lines) or *full interaction* (solid lines) and of *no interaction*, considered as the baseline. The results are split into sub-figures to show the effect of the season in which the flood takes place and of the farm coping tactic. The red lines correspond to the case in which the winery building is flooded, and the blue lines to the case in which it is not flooded (safe).

### 4.2.1 Qualitative analysis

Figure 7 shows two types of implications of the assumptions A1 to A3. The first type of implication results from the fact that, when all interactions are not taken into consideration, the extent of some indirect damage cannot be captured, leading to underestimation of damage in cases with *no interaction* and *partial interaction* compared to the case with *full interaction*. When the winery building is safe, this applies in all seasons: the blue solid lines (*full interaction*) are always
above the blue dashed lines (*partial interaction*), which are above 0 (*no interaction*). When the winery is flooded, the aforementioned underestimation also applies in all seasons except autumn: the red solid lines are always above the red dashed lines, which are above 0, except in autumn where the red solid lines are below 0.

The second type of implication occurs in autumn, when assumption A2 leads to some double counting, and hence to overestimation of damage in the cases of *no interaction* and *partial interaction* compared to the case with *full interaction*.
Wine production depends on the yield of grapes supplied by the farms, and hence on the grape losses incurred by the

| | Impact | | Winter | | Spring | | Summer | | Autumn | |
|---|---|---|---|---|---|---|---|---|---|---|
| | | | Min | Max | Min | Max | Min | Max | Min | Max |
| Winery safe | Absolute* | Total | 4,800 | 330,000 | 66,928 | 1,792,658 | 54,288 | 1,277,456 | 48,531 | 1,152,458 |
| | | Direct | 4,800 | 330,000 | 22,931 | 1,028,886 | 17,564 | 598,974 | 4800 | 449,190 |
| | Relative** | Total | 0.006 | 0.415 | 0.084 | 2.258 | 0.068 | 1.609 | 0.061 | 1.451 |
| | | Direct | 0.006 | 0.415 | 0.028 | 1.295 | 0.022 | 0.754 | 0.006 | 0.566 |
| Winery flooded | Absolute* | Total | 5,742,720 | 6,067,920 | 6,368,860 | 8,087,108 | 3,533,654 | 4,746,838 | 6,350,297 | 7,444,240 |
| | | Direct | 3,484,800 | 3,810,000 | 3,502,931 | 4,508,886 | 3,497,564 | 4,078,974 | 3,484,800 | 3,929,190 |
| | Relative** | Total | 7.232 | 7.642 | 8.021 | 10.185 | 4.450 | 5.978 | 7.997 | 9.375 |
| | | Direct | 4.388 | 4.798 | 4.412 | 5.678 | 4.405 | 5.137 | 4.389 | 4.948 |

\* Measured in euros (€)

\** Measured in *number of times the annual potential added value (PAV) of the CWS* (which is € 794,000): e.g. the minimum amount of direct impact of a flood in summer when the winery is flooded represents 4.405 times the PAV.

*Remark:* Min and Max refer to the impact issued from the COOPER model for simulations of, respectively, the smaller-scale and the larger-scale flood events

**Table 5.** Comparative of total and direct impacts issued from the COOPER model for the baseline scenario. Detail by season, scale of flood event (min and max) and winery position

farms. Under assumption A2, (*partial interaction* and *no interaction*), wine production in the cooperative winery in autumn is independent of the losses incurred by the farms. Thus, under assumption A2, the part of the harvest that is lost to the farms is also considered lost to the cooperative winery. The bigger the flood, the bigger the losses to the farms, the more the double counting. In other seasons, no such double counting occurs because the quantity of grapes in the winery building does not depend on the quantities currently present on the farms. For instance, in winter, wine production in the winery building depends on the grapes harvested in autumn, not on the grapes currently growing in plots that will be harvested the following season.

Coming back to our explanation for underestimation of damage in the other cases, whether or not the winery building is flooded has no impact on the sign of the differences, even if the magnitude is much greater when the winery building is not flooded. This difference in magnitude originates from the fact that material damage is much greater when the winery building is damaged, and the relative difference is consequently lower.

In spring and in summer, there are differences between the cases of *partial interaction* and *no interaction* but the differences are smaller than between the cases of *full interaction* and *partial interaction*. It is assumption A3 that leads to the following statement: In cases of *partial interaction* and *no interaction*, the costs of winemaking in the year following

the flood are overestimated insofar grape losses at the farms level are not taken into account in the cost estimation. This also happens in autumn.

In winter, there are no losses of grape yields in flooded plots. Grape losses in this season only occur to farms that apply the internal tactic when their building is flooded insofar such tactic provokes further yield damage due to task misperformance. Observable differences are explained by assumption A1, which is also the reason why the differences

begin at flood extent 30 (first building impacted). The fact that the difference between cases of *partial interaction* and *no interaction* is noticeable in winter is related to the importance of the seasonal tasks performed in terms of loss of yield. This is also the case in autumn, but not in spring and summer. In spring and summer, the tasks are less important with respect to the future yield, and the plots are also more vulnerable: grape yield losses are more directly linked to flooding of plots than to flooding of the farm building. In the case of the external tactic, it is also assumption A1 that explains the

difference, but the impact is not loss of the grape yield, but increments in winegrowing costs. In Figure 7, this increment is important only in winter. In this case (external tactic, winter), in Figure 7, the curves for *partial interaction* and *full interaction* match perfectly.

### 4.2.2   Quantitative analysis

First, when the winery building is flooded, the differences increase with the extent of the floods, but remain negligible,

except in autumn. This because the material damage to the winery building is very severe. In autumn, double damage accounting leads to a difference of between 10% and 20%, increasing linearly with the number of plots flooded, and hence with the extent of the floods (because of the spatial configuration chosen, see Figure 4).

When the winery building is not flooded, in spring, summer, and autumn, the differences are about 10% (which, in absolute terms, ranges between € 6,000 to more than € 170,000, depending on the magnitude of the flood), increasing

to 20% in autumn for the internal tactic when the farm building is flooded. In winter, the differences are negligible as long as no farm building is flooded. Otherwise it is about 20% (€ 66,000 in the biggest event) for the external tactic and 40% (€ 132,000 for the biggest flood) for the internal tactic.

It is important to note that in spring and summer, the differences between *partial interaction* and *full interaction* are bigger than between *no interaction* and *partial interaction*, independently of the chosen tactic. This is also the case in

autumn, but only for the external tactic. This means that in these cases, it is more important to clarify the links between economic entities (farms and the winery) than within economic entities (plots and the farm building). In winter, for both tactics, and in autumn for the internal tactic, there is a clear difference between the three cases. The gap between the *no interaction* and *partial interaction* is bigger than the gap between the *partial interaction* and full interaction. Consequently, in these cases, it is more important to establish the links within economic entities (between material components) than

between economic entities. Finally, as floods can occur in any season, it is impossible to draw final conclusions about which type of interactions it is most important to take into consideration. Both should to be taken into account.

### 4.3 Influence of configurations of interactions in damage estimation

The analyses presented in section 4.2 apply to a particular configuration of interactions. All the farms own exactly 10 plots (homogeneous size) and have more or less the same ratio of plots located in the flood prone area (homogeneous exposure). In the case of *no interaction*, it is not important to know exactly which farm the plots belong to: as explained previously, in this case, when assessing damage, it is assumed that the farm to which a plot belongs is not flooded. This is not the case for *partial interaction*, or for *full interaction*. In these cases, even if all the material components are located at exactly the same place, the way they are linked may have an influence on flood damage. In this section, we analyze this influence.

In section 4.2 we also showed that interactions have the most influence when the cooperative winery is not flooded, so in this section, we detail the case when the cooperative winery is not flooded (Figure 8). However, in the spirit of full disclosure, we also briefly analyze the case when the cooperative winery is flooded with no additional figures. In this case, the relative differences between the configurations are very similar. In fact, the main damage originates in the cooperative winery, and any difference originating from farm heterogeneity is offset at the level of the cooperative winery. This has direct implications for the significance of the double counting bias mentioned in the previous section: it is almost independent of farm heterogeneity (about $12\%$ in the case of *no interaction*). This is also true for other seasons for which the damage propagation bias is negative, but almost negligible ($1-2\%$ in spring, $1-3\%$ in summer, $0-2\%$ in winter).

#### 4.3.1 Qualitative analysis

When the cooperative winery is not flooded, Figure 8 shows the relative differences in damage at the system level between the simulations of configurations presented in Table 4 for the case of *full interaction*, compared to the case of *no interaction*.

First, it can be seen that in all seasons, there is always less damage in the case of *no interaction* than in the case of the *full interaction*, for all configurations of links. The same bias as in section 4.2 is observed in spring, summer and autumn: there is a positive difference of about $10\%$ between simulations with *full interactions* and simulations with *no interactions*. Differences between the configurations of links appear when the first farm building is flooded (flood of extent 30) and become more visible in parallel with the increase in flooded buildings.

The *size* configuration (green line), which represents big farms and small farms with comparable exposure, does not introduce a major difference from the *homogeneous* configuration (black lines) in which all farms are the same size with equivalent exposure. This is also true for the two configurations that introduce heterogeneity in terms of exposure: *exposure-best* (solid blue lines) and *exposure-worst* (dashed blue lines).

Clear differences only appear when both types of heterogeneity are introduced and combined. In this case, the configuration that suffers the most damage is always the *size-exposure-worst* one (dashed red lines). In this configuration, all the buildings belonging to the big farms are located in the flood-prone area, but their plots are located outside. When their building is flooded, all the tasks required for their production are disrupted, which results either in extra costs (external

tactic) or extra yield losses (internal tactic). This is the worst configuration for such effects. The configuration that suffers the least damage is always the *size-exposure-best* one (solid red lines). In this configuration, all the big farms are located outside the flood-prone area. Thus, the buildings located in the flood-prone area belong to the small farms, and potential disruption of tasks only concerns a few plots. As these plots are located in the flood-prone area and suffer direct damage from the flood, the disruption of tasks is not that important. These differences are particularly clear in winter when many

of the tasks on plots have to be completed under both tactics, and in autumn under the internal tactic, when being unable to harvest involves high yield losses.

### 4.3.2 Quantitative analysis

Concerning the magnitude of the differences between configurations, relative differences may be quite important. Under the configuration that suffers the most damage (*size-exposure-worst*), relative differences may be close to 110% in winter

(up to €360,000), 60% in autumn (as far as €690,000), 20% in summer and spring (up till €250,000 and €360,000) under the internal tactic, decreasing to 60% in winter (down to €198,000), 20% in summer, 10% in autumn and spring (up to €130,000) under the external tactic. Under the configuration that suffers the least damage (*size-exposure-best*), relative differences may be close to 20% in winter (€66,000 for the worst flooding events), about 5% in the other seasons (between €90,000 and €60,000 depending on the season) under the internal tactic, decreasing to 10% in winter, and

about 5% in all seasons under the external tactic.

Compared to the results in the previous section, it is clear that the configuration of links matters for quantitative analysis. To grasp whether a difference is significant, the two sources of heterogeneity need to be combined: in terms of the size of farms and in terms of plot exposure.

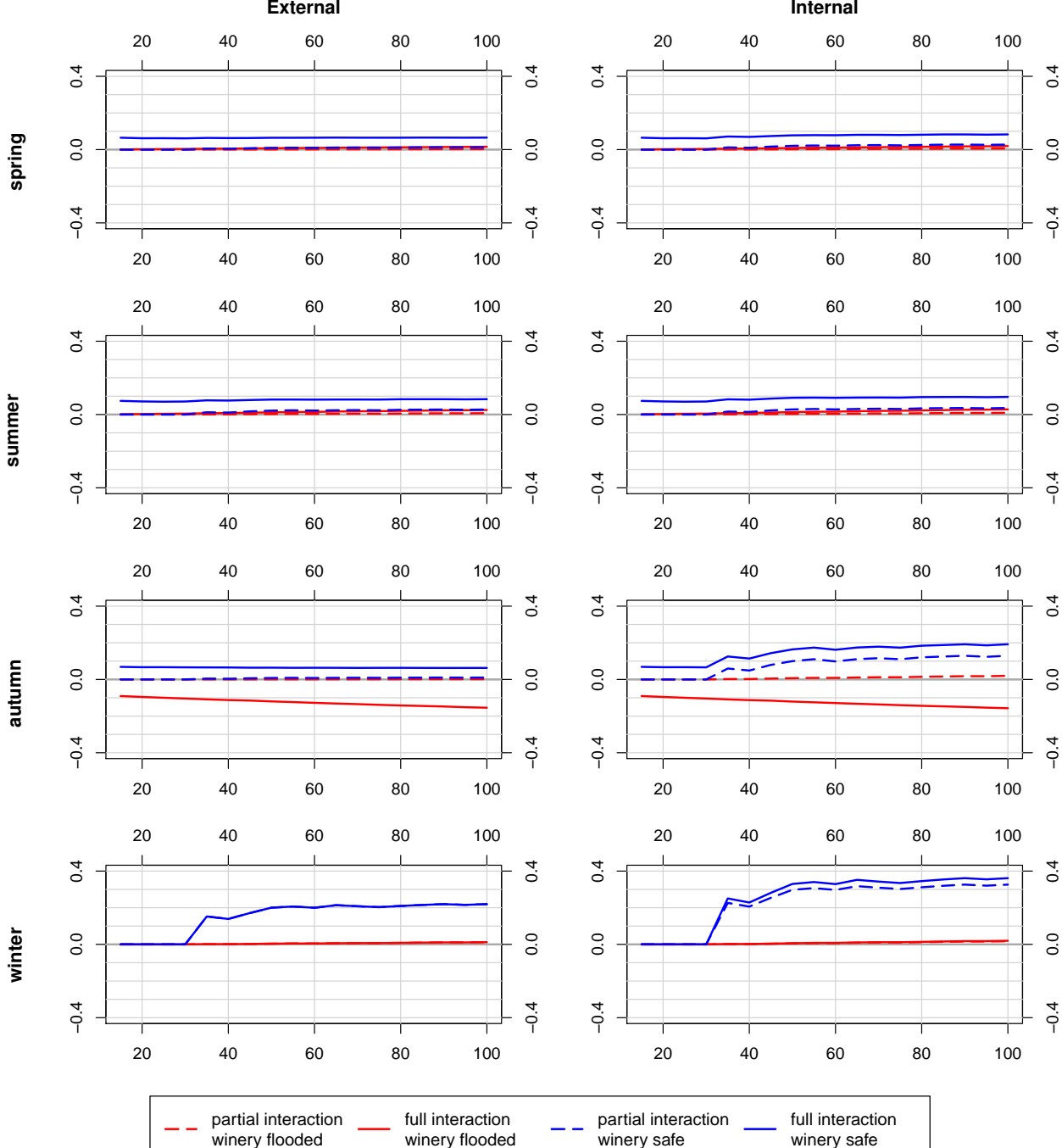

**External**   **Internal**

spring

summer

autumn

winter

partial interaction
winery flooded

full interaction
winery flooded

partial interaction
winery safe

full interaction
winery safe

*Remark*: In each figure, the x-axis shows the extent of the flood; the y-axis corresponds to the relative difference in the quantification of flood impacts compared to the baseline simulation (*no interaction*). Each column represents a different farm coping tactic.

**Figure 7.** Implications of the level of interactions taken into account for damage assessment (homogeneous case)

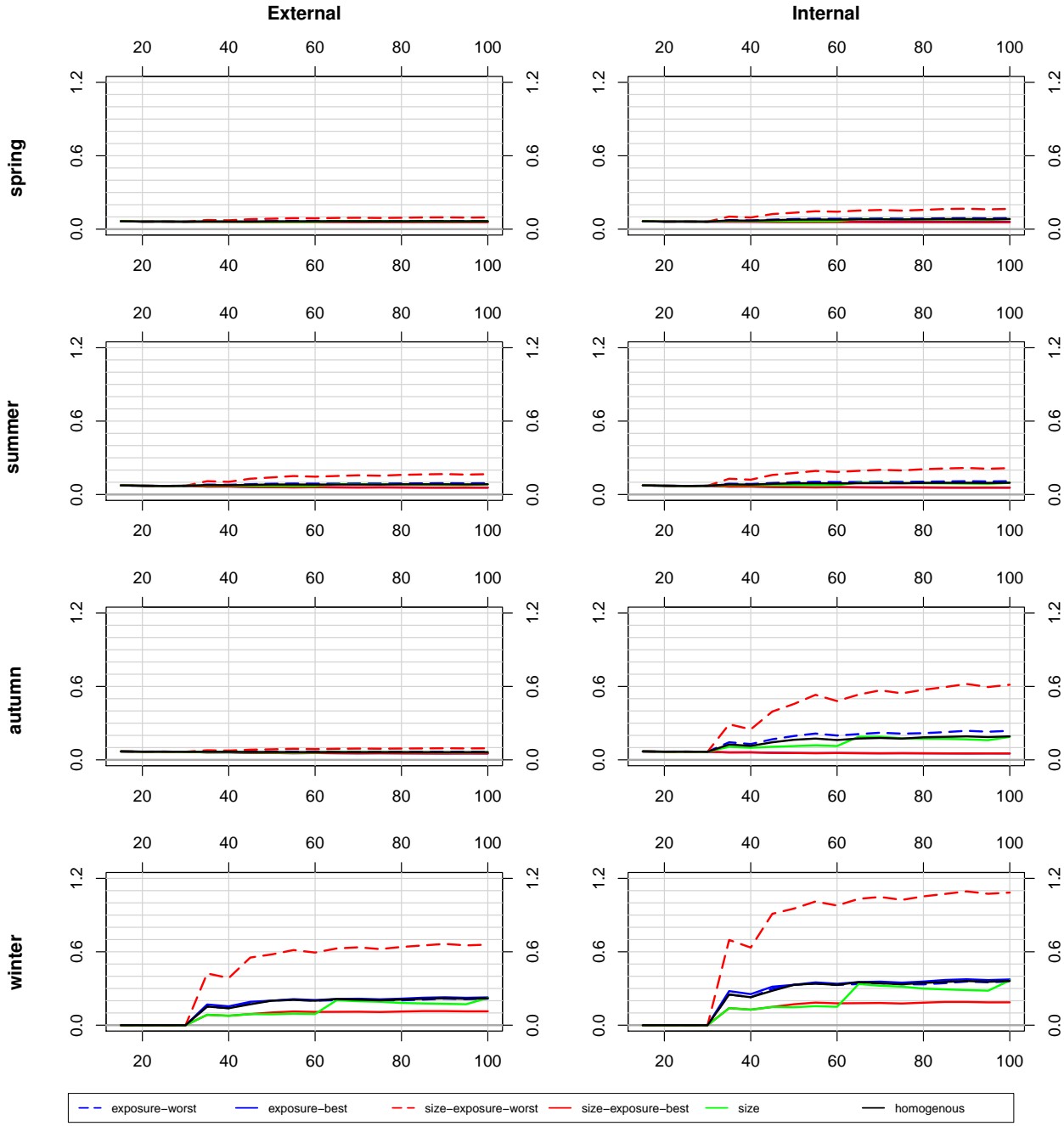

*Remark*: In each figure, the x-axis indicates the extent of the flood; the y-axis corresponds to the relative difference in the quantification of flood impacts compared to the baseline simulation (*no interaction*). Each column represents a different farm coping tactic.

**Figure 8.** Implications of the configuration of the interactions for damage assessment (Winery safe)

## 5 Discussion

### 5.1 Importance of interactions in damage assessment

Current damage assessment at local level within complex productive systems considers agents and their material entities separately but does not include the links between them. Our experiments show that this kind of practice can involve two types of bias. The first bias, i.e. the misrepresentation or absence of links, leads to underestimation of flood impacts due to inherent inaccuracy in the spreading of disturbances within the system. The second bias was less expected and is in contradiction with the arguments put forward by Penning-Rowsell et al. (2013b). It overestimates damage when failing to take interactions into account. The origin of such a bias can be traced back to the fact that, even when entities are considered independently, their schedules are not taken into account. As a result, some material components could be considered to be present at two places at once, thereby leading to double accounting of some material damage.

These two types of bias can be extrapolated to other types of economic systems. Indeed, systems in which the substitution of inputs is not plausible would face the same problems in the estimation of flood damage if the interactions between the component entities are not taken into account. For instance, this may be the case of systems organized like the CWS, in which input substitution is not permitted by the nature of the product, by rigidities introduced by contracts, or by the lack of substitutes for very specific goods, as observed in the automobile and electronic industries after the 2011 flood in Thailand (Haraguchi and Lall, 2015). The second type of bias will be found in any system in which material components move through different economic entities (basically the case in all supply chain systems). When there is no clear idea of the location of the product, and hence no thorough understanding of the production processes and schedules, there is a high probability of overestimating economic damage due to duplicate entries in an inventory.

Our experiments also showed that if interactions are to be taken into account, they must be thoroughly characterized. In such regard, results in section 4.3 highlight the importance of the configuration of links between material components. These results are particularly relevant to the extent that location (thus spatial exposure), the vulnerability of individual equipment, and the rules governing links between material entities were the same across simulations. Under such conditions, current flood damage assessment would find no difference between configurations, even though the different configurations of links between entities lead to different damage intensities. To fail in properly characterizing the existing interactions within a system can lead to bias in the resulting flood damage estimation, compromising the very advantage of taking interactions into account.

The described improvements in flood damage estimation come nonetheless at a cost in terms of information gathering. A thorough characterization of the interactions present in a system as the CWS can be highly resource-consuming. In that regard, the comparison made between our so-called *within activity* and *between activity* interactions does not enable us to judge whether some types of interactions are more important than others. The results obtained (section 4.2) show that the importance of the type of interactions depends on the season, and consequently, on the underlying production processes. Furthermore, it appears that concerning productive units composed of elements of very different nature in different locations –as the CWS' farms– taking both types of interactions into account is highly recommended, whereas

for productive units whose means of production are concentrated in one place –e.g. the CWS' winery– the characterization of the *between activities* interactions may suffice (thus, assuming that all elements in those productive units are equally affected by a flood).

## 5.2 Contribution of a computational laboratory

Finally, we would like to highlight that our method proposes –and is based on– a computational laboratory for flood damage assessment. It enables the estimation of damage to a CWS originating from small-, medium-, or large-scale flood events. While we did not use the same modeling approach as Koks et al. (2014), like them, we consider that, as impact mechanisms differ depending on the scale of the event, this wide view has undeniable advantages over the study of a single phenomenon. For instance, our results clearly show that, at least in the case of CWSs, contrary to what is claimed, it is not appropriate to use *approaches that calculate production losses using a fixed share of direct damage* (Meyer et al., 2013) for all types of events. Although this article focuses on a CWS, the development procedure is applicable to other CPS. In this sense, the contents of this article, together with the information in Nortes Martinez et al. (2019), can be used as guidelines for the development of COOPER-like models applied to other CPS.

## 5.3 Limits to the study

Our analysis presents several limits that should be considered.

First, like in all modeling approaches, we have simplified some of the processes. In the present version of the COOPER model, the behavior of economic entities is representative of what we encountered in our field surveys and in past research. Economic entities show reactive behavior, i.e. they try to return as quickly as possible to the *statu quo ante*: They repair each damaged material component and whenever possible, respect the normal production process. "Real life" cases also include agents with more planned behavior, whose decisions will depend, for instance, on the level of damage incurred, their financial situation when a flood occurs, and their business plans. Moreover, agents may use tactics to actively deal with floods (e.g. moving vulnerable equipment), or production disruption (e.g. in the case of wineries, borrowing/renting external equipment to enable wine production) that are beyond the scope of this article but could have an impact on our results. The impact that different agents' behaviors can have on our results and on the response of the whole system to floods constitutes a future line of research that merits attention.

Second, as mentioned above, we focussed on two specific kinds of interactions within the boundaries of the system. Other interactions observed in real cases concerning farm cooperation and organization – e.g. equipment and/or labor sharing, solidarity after flood events – or farm-winery cooperation – e.g. bilateral help in the case of flooding – are not incorporated in the current version of the model. The impacts of those interactions are thus beyond the scope of the present article. The impact they may have on our results nonetheless deserves further investigation and is consequently a potential line of research. Similarly, the interactions between the CWS and other entities – e.g. input/equipment providers, sellers, insurers, or banks – are also outside the purview of this article, but their effects also merit further investigation.

Finally, we chose a CPS that is organized like a star, with a central element. While appropriate for the CWS, this representation does not fit some economic sectors that would be better represented by a multi-node system, or even a no-node system.

## 6   Conclusion

Although left aside in current practice, the introduction of explicit interactions in productive systems has a non-negligible
impact on the amount of damage estimated at a microeconomic scale. The characterisation of these interactions requires the introduction of links between the material components mobilised by the productive system, the nature of which depends directly on the tasks and operations necessary for the production process. Not taking these interactions into account can lead to two opposite effects: an underestimation of damage if the propagation of disturbances is poorly represented, and an overestimation of damage if the location of the product is poorly represented. This observation does not
allow us to give a general recommendation, one way or the other, to correct the estimates currently made.

      The effort required to represent these interactions, compared to current practice, is quite substantial. It includes the acquisition of additional information, which so far has been seldom used in the definition of damage functions. It also involves a real effort to understand the production processes, which is in any case greater than what is required to produce the current damage functions.

This observation may appear to be a barrier to improving the operational practice of damage estimation, particularly in the context of economic evaluations such as cost-benefit analyses based on avoided damage, carried out on a regional or national scale. However, we believe that it deserves particular attention at a local scale. We are thinking in particular of projects that could have negative consequences in terms of exposure to flooding, which would have to be compensated for. In this case, an estimate that takes into account the understanding of the production system and its spatial exposure
seems necessary to better establish any monetary estimate of compensation, and to introduce confidence with the over-flooded stakeholders. We are also thinking of all projects that aim to reduce the vulnerability of productive systems. In this respect, our approach could provide an evaluation framework and lead to a better understanding of the vulnerability of such systems.

      In terms of perspectives, our approach, particularly in its virtual laboratory component, can make it possible to ex-
plore combinations of phenomena for which it is difficult to obtain empirical data, such as, for example, the impact of a succession of floods in close proximity, the conjunction of a flood with a market-type hazard affecting the modelled sector.

      Our approach could also be complemented by an analysis of the resilience of the activities, by following the financial impact of floods over time and checking whether the financial situation of the activities is compatible with the continuation of their activities, or if on the contrary it seems to lead to bankruptcy.This questions a key assumption of economic
analyses based on avoided damage, which implicitly assumes that the damage caused by floods is estimated, without the disappearance of the stakes that have suffered them. The relevance of this assumption should also be discussed in future work.

Finally, our approach could be combined with a practice of observing the impacts with the actors directly concerned. Indeed, although our analyses are based on in-depth surveys of winegrowers and managers of cooperative wineries, we can only benefit from discussing both our results and our modelling approach with the main stakeholders. This discussion could make it possible to clarify any overly strong assumptions that we might have made. It could also make it possible to give the virtual experience of events that the interested parties would not have experienced, in order to be better prepared for them. This is the ambition of a process that our team has started within the framework of the so-ii flood impact observatory (so-ii.org), with the establishment of a long-term partnership with people particularly concerned by floods.

*Code and data availability.* The COOPER model and the data needed to perform the present analysis are available online at the CoMSES computational model library (https://www.comses.net/codebases/6038/releases/1.0.1/).

*Author contributions.* DNM, FG, and JR developed COOPER model, DNM and FG performed the analysis and drafted the first version of the manuscript. All authors discussed the results and edited the manuscript.

*Competing interests.* The authors declare that they have no conflict of interest.

*Acknowledgements.* We acknowledge financial support from the French Ministry of Environment, trough the RDT research program in the framework of the Retina project 13-MRES-RDT-2-CVS-023, and through grant decision no. 2102897179. David Nortes Martínez received a doctoral fellowship from the French national research institute for agriculture, food and the environment (INRAE).

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

## Appendix A: Mathematical appendix

### A1   Cost-revenue sharing rule

In the COOPER model (Nortes Martinez et al., 2019), to split cost and revenues among associated farmers, the cooperative winery proceeds according the following rule (Touzard et al., 2001; Biarnès and Touzard, 2003; Jarrige and Touzard, 2001):

$$TC_i = \left( \frac{F + V}{\sum_{i=1}^{n} q_i} q_i \right) \quad (i = 1, 2...n) \tag{A1}$$

$$B_i^o = pq_i - TC_i = pq_i - \left( \frac{F + V}{\sum_{i=1}^{n} q_i} q_i \right) \quad (i = 1, 2...n) \tag{A2}$$

Where:

- $TC_i$ is the share of the wine-making cost in the winery for the farm i.

- $B_i^o$ is the share of the profit in the winery for the farm i.

- $pq_i$ is the share of revenue of the farm i.

- $\frac{F+V}{\sum_{i=1}^{n} q_i} q_i$ is the decomposed wine-making cost in the winery for the farm i.

  - $F$ is the fixed (structural) wine-making cost.

  - $V$ is the variable (operational) wine-making costs.

  - $\sum_{i=1}^{n} q_i$ is the total production in the cooperative winery, as a sum of the individual productions of the associ-
ated farms.

  - $q_i$ is the production of the farm i.

## A2   Imputation of winery's reparation costs among associated farmers

As we stated in section 3.2, when the cooperative winery is flooded we can differentiate two mechanisms for imputing reparation costs shares to associated farmers. The first mechanism imputes costs proportionally to the farmers' individual
yields as in equation A3

$$R_i = \left( \frac{R}{\sum_{i=1}^{n} q_i} q_i \right) \tag{A3}$$

Where:

1. $R_i$ is the reparation costs imputed to farm i

2. $R$ is the total monetary value of reparations

3. $\sum_{i=1}^{n} q_i$ is the total production in the cooperative winery, as a sum of the individual productions of the member
     farms.

4. $q_i$ is the production of the farm i.

The second mechanism imputes costs proportionally to the number of farmers because the CWS' production is lost. This mechanism comes formalized in equation A4

$$CT_i = \frac{R+F}{N} \tag{A4}$$

Where:

1. $CT_i$ is the total cost imputed to farm i

2. $F$ is the monetary value of the fixed vinification costs

3. $R$ is the total monetary value of reparations

4. $N$ is the number of farms members in the cooperative winery

### A3 Hierarchy of impacts

Inasmuch as the flood impacts over the different material entities can be simultaneous, the effects can be summed. However, to avoid problems related to double accountability and, also, to be able to trace each impact back to its origin we have chosen to introduce hierarchic levels over flood impacts.

Flowchart A1 sketches out the hierarchy levels by entities. Before we can analyze it, we need to introduce the following new nomenclature and definitions: For each productive plot $\gamma_\kappa$, owned by farm $i$, we can express its yield as

$$q_{i_T\kappa} = q_{i\kappa} + q_{i_D\kappa} \tag{A5}$$

Where:

1. $q_{i_T\kappa}$ is the potential yield in plot $\gamma_\kappa$ of farm $i$

2. $q_{i\kappa}$ is the effective yield in plot $\gamma_\kappa$ of farm $i$

3. $q_{i_D\kappa}$ is the damaged yield in plot $\gamma_\kappa$ of farm $i$ by the flood

The term $q_{i_D\kappa}$ "stores" the total of yield damaged, whether its origin is in the direct submersion of the yield or provoked by vine damages.

In our system, each farm $i$ owns a number $n_i$ of plots. Aggregating all those plots, each farm $i$ owns a total extent $\Gamma_i$ that can be expressed as:

$$\Gamma_i = \sum_{\kappa=1}^{n_i} \gamma_{i\kappa} \tag{A6}$$

Using equation A6, we can express equation A5 at farm level as:

$$\sum_{\kappa=1}^{n_i} q_{i_T\kappa} = \sum_{\kappa=1}^{n_i} q_{i\kappa} + \sum_{\kappa=1}^{n_i} q_{i_D\kappa} \tag{A7}$$

Where:

1. $\sum_{\kappa=1}^{n_i} q_{i_T\kappa}$ is the potential yield of farm i

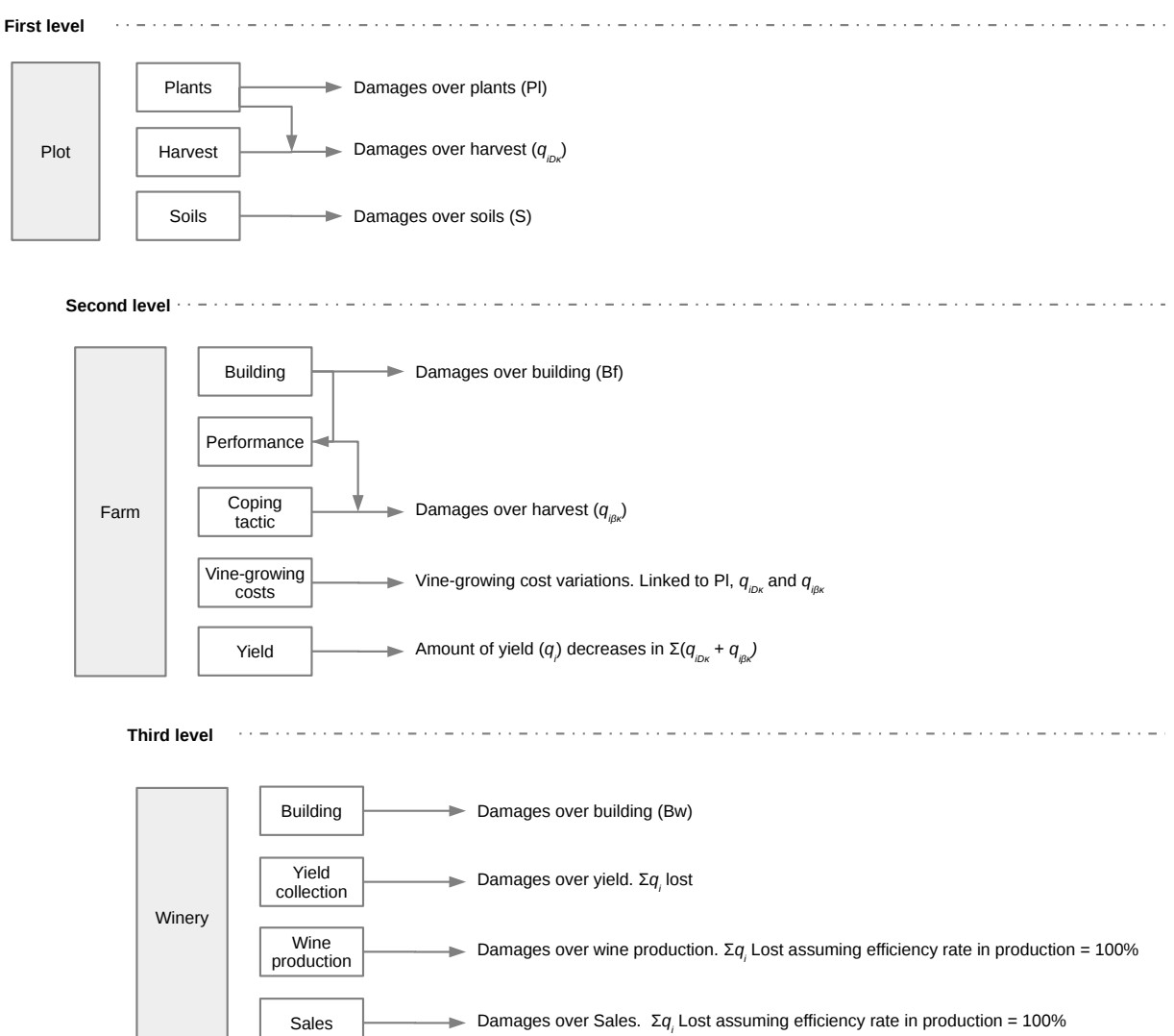

**Figure A1.** Hierarchy of damages for a flood hitting entities altogether

2. $\sum_{\kappa=1}^{n_i} q_{i\kappa}$ is the effective yield of farm i

3. $\sum_{\kappa=1}^{n_i} q_{i_D\kappa}$ is the damaged yield of farm i

And the term $\sum_{\kappa=1}^{n_i} q_{i_D\kappa}$, as in the individual case, "stores" the total of yield damaged, whether its origin is in the direct submersion of the yield or provoked by vine damages.

At the same time, we know that, depending on the coping strategy the farm adopts, we can have additional damages over the yield. To take such effect into account, and, therefore, know the real value of $\sum_{\kappa=1}^{n_i} q_{i\kappa}$, we need to modify equation A5 introducing the new term, $q_{i_\beta\kappa}$:

$$q_{i_T\kappa} = q_{i\kappa} + q_{i_D\kappa} + q_{i_\beta\kappa} \tag{A8}$$

Where:

1. $q_{i_T\kappa}$ is the potential yield in plot $\gamma_\kappa$ of farm $i$

2. $q_{i\kappa}$ is the effective yield in plot $\gamma_\kappa$ of farm $i$

3. $q_{i_D\kappa}$ is the damaged yield in plot $\gamma_\kappa$ of farm $i$ by the flood

4. $q_{i_\beta\kappa}$ is the damaged yield in plot $\gamma_\kappa$ of farm $i$ caused by the coping strategy of the farm $i$

Then equation A7 becomes:

$$\sum_{\kappa=1}^{n_i} q_{i_T\kappa} = \sum_{\kappa=1}^{n_i} q_{i\kappa} + \sum_{\kappa=1}^{n_i} q_{i_D\kappa} + \sum_{\kappa=1}^{n_i} q_{i_\beta\kappa} \tag{A9}$$

Where:

1. $\sum_{\kappa=1}^{n_i} q_{i_T\kappa}$ is the potential yield of farm i

2. $\sum_{\kappa=1}^{n_i} q_{i\kappa}$ is the effective yield of farm i

3. $\sum_{\kappa=1}^{n_i} q_{i_D\kappa}$ is the damaged yield of farm i

4. $\sum_{\kappa=1}^{n_i} q_{i_\beta\kappa}$ is the damaged yield of farm $i$ caused by the farm i's coping strategy

Or alternatively,

$$q_{i_T} = q_i + q_{i_D} + q_{i_\beta} \tag{A10}$$

Where:

$$q_{i_T} = \sum_{\kappa=1}^{n_i} q_{i_T\kappa} \qquad q_i = \sum_{\kappa=1}^{n_i} q_{i\kappa} \qquad q_{i_D} = \sum_{\kappa=1}^{n_i} q_{i_D\kappa} \qquad q_{i_\beta} = \sum_{\kappa=1}^{n_i} q_{i_\beta\kappa} \tag{A11}$$

Up-scaling a level in the production chain, we can express the amount of yield provided as input to the cooperative winery, $Q_w$, as the aggregation of the individual yields of its associates:

$$Q_w = \sum_{i=1}^{n} q_i = \sum_{i=1}^{n}\sum_{\kappa=1}^{n_i} q_{i\kappa} \tag{A12}$$

Where $n_i$ is the number of plots, $\gamma_\kappa$, of farm i, and $n$ is the number of farms

Returning to flowchart A1, we can use the new nomenclature to clearly scout damages when different entities are flooded at the same time. Let us assume i) the flood hits the system in year $t = 1$, and ii) seasonal sequence is winter-spring-summer-autumn. Then, if the flood hits the system in:

1. **Winter**. Impacts over plots flooded are reduced to reconditioning of soils (S)

   Impacts over farms flooded include buildings (B1) and performance. If opting for *external*, $q_{i_\beta\kappa} = 0$, in each plot owned by flooded farms. Therefore in autumn, when harvest is done, in each productive plot owned by those farms $q_{i\kappa} = q_{i_T\kappa}$, thus $q_i = q_{i_T}$ at farms level for $t = 1$. If opting for *internal*, $q_{i_\beta\kappa} > 0$, in each plot owned by flooded farms, so in autumn $q_{i\kappa} < q_{i_T\kappa}$ in each plot owned by flooded farms, and $q_i < q_{i_T}$ at farms level for $t = 1$. In any case, vine-growing cost will vary

   Impacts over wineries incorporate damages over buildings (B2) and performance. It will make the system lose $Q_w$ of $t = 0$, but will have no effect over $Q_w$ of $t = 1$. Since $Q_w$ is lost, there will be no revenues for farms in $t = 1$, and the ones expected in $t = 2$ will be linked to the farms coping tactic. Wine-making cost will vary reflecting both situations.

2. **Spring**. Impacts over plots flooded include reconditioning of soils (S), losses of yield $q_{i_D\kappa} > 0$ and vine destruction (Pl)

   Impacts over farms flooded include buildings (B1) and performance. If opting for *external*, $q_{i_\beta\kappa} = 0$, in each plot owned by flooded farms. Therefore in autumn $q_i < q_{i_T}$ in the amount given by $q_{i_D}$ at farms level for $t = 1$. If opting for *internal*, $q_{i_\beta\kappa} > 0$, therefore in autumn $q_i < q_{i_T}$ too, but in the amount $q_{i_D} + q_{i_\beta}$. As in winter, vine-growing-cost will vary

   Impacts over wineries are the same as in winter. Since in spring destruction of vines is likely to happen, the impacts over wine-making costs and revenues can last longer in time.

3. **Summer**. Impacts over plots and farms are the same as exposed for spring, while impacts over wineries are reduced to reparation costs over buildings and materials (B2). Impacts over revenues and wine-making cost in

$t = 2$ —and potentially further in time— will reflect the level of destruction in plots and the coping tactics chosen by farms.

4. **Autumn**. Impacts over plots and farms are the same as exposed for spring. Impacts over wineries comprise damages over buildings (B2) and performance. It will make the system lose $Q_w$ of $t = 1$.

As we can see, in $t = 1$ eventually all production gets lost, but for several reasons:

- It exists $q_{i_D\kappa} > 0$ at each flooded plot. Therefore at system level we have $\sum_{i=1}^{n} \sum_{\kappa=1}^{n_i} q_{i_D\kappa} > 0$ provoked by the direct impact of floods over plots

- If farm's coping tactic is *external*, then $q_{i_\beta\kappa} = 0$. There is no added damage by the farm, and the yield lost by the winery is:

$$Q_w = \sum_{i=1}^{n} \sum_{\kappa=1}^{n_i} q_{i_T\kappa} - \sum_{i=1}^{n} \sum_{\kappa=1}^{n_i} q_{i_D\kappa} \tag{A13}$$

- If farm's coping tactic is *internal*, then $q_{i_\beta\kappa} > 0$, the added damage by each farm is $\sum_{\kappa=1}^{n_i} q_{i_\beta\kappa}$, and the yield lost by the winery is

$$Q_w = \sum_{i=1}^{n} \sum_{\kappa=1}^{n_i} q_{i_T\kappa} - \sum_{i=1}^{n} \sum_{\kappa=1}^{n_i} q_{i_D\kappa} - \sum_{i=1}^{n} \sum_{\kappa=1}^{n_i} q_{i_\beta\kappa} \tag{A14}$$

Revenues in $t = 2$ will be null and wine-making cost will be reduced to the winery's fixed (structural) cost. Due to vine destruction at plot level, as it happens in spring and summer, effects over revenues and wine-making cost are expected to last longer in time, reflecting such vine destruction.

## A4 Impact calculation

Table A1 offers a more insightful overview to the calculation of impacts by comparison of a *Business as Usual Scenario* (BAU) with a *Flood Scenario* (FS).

As the reader can appreciate, impacts can be calculated at collective level –that is, the whole CWS– and at individual level –for each farm $i$ associated to the CWS– at any moment $t$, where $t$ represents the model's timesteps, namely the number of seasons (not the number of years). It is worth noting that $q'_{i,t} - q_{i,t}$ in table A1 is not the same as $q_{i_D}$ in equation A10. In the equation, we refer only to the yield damaged by the flood, while $q'_{i,t} - q_{i,t}$ also includes the yield lost because of disability of an agent to perform an assigned task due to the flood. That is to say, it includes $q_{i_\beta}$ and $Q_\omega$. Aggregating the different components in table A1, we obtain the total impact for each individual farm $i$ and the whole CWS as shown in equations A15 and A16:

$$Imp_{i,t} = (I'_{i,t} - I_{i,t}) + (p + v_{vg_i} + v_{wm})(q'_{i,t} - q_{i,t}) + F_{wm}\frac{\sum_{i=1}^{n} q_{i,t} - \sum_{i=1}^{n} q'_{i,t}}{\sum_{i=1}^{n} q'_{i,t} \sum_{i=1}^{n} q_{i,t}} \tag{A15}$$

| Variable | Impact ($Imp_t = \boldsymbol{FS}_t - \boldsymbol{BAU}_t$) | |
| | Farm $i$ | CWS |
| --- | --- | --- |
| $I_t$ | $I'_{i,t} - I_{i,t}$ | $I'_t - I_t$ |
| $Q_t$ | $p(q'_{i,t} - q_{i,t})$ | $p\left( \sum_{i=1}^{n} q'_{i,t} - \sum_{i=1}^{n} q_{i,t} \right)$ |
| $Cvg_t$ | $v_{vg_i}(q'_{i,t} - q_{i,t})$ | $v_{vg_i}\left( \sum_{i=1}^{n} q'_{i,t} - \sum_{i=1}^{n} q_{i,t} \right)$ |
| $Cwm_t$ | $v_{wm}(q'_{i,t} - q_{i,t}) + F_{wm} \frac{\sum_{i=1}^{n} q_{i,t} - \sum_{i=1}^{n} q'_{i,t}}{\sum_{i=1}^{n} q'_{i,t} \sum_{i=1}^{n} q_{i,t}}$ | $v_{wm}\left( \sum_{i=1}^{n} q'_{i,t} - \sum_{i=1}^{n} q_{i,t} \right)$ |

*Remark:* $C_{vg_{i,t}} = F_{vg_i} + v_{vg_i} q_{i,t} \mid C_{wm_t} = \frac{F_{wm}}{\sum_{i=1}^{n} q_{i,t}} + v_{wm} q_{i,t}$

*Key:* $I_t$ = Investment | $Q_t$ = Production | $C_{vg}$ = Vine-growing cost | $C_{wm}$ = Wine-making cost | $q_{i,t}$ = yield of farm $i$ at moment $t$ |

$\sum_{i=1}^{n} q_{i,t}$ = sum of yields of all farm $i \in [1, n]$ at moment $t$, where $n$ = number of farms in CWS | $p$ = market price of wine |

$v_{vg_i}$ = variable vine-growing cost farm $i$ | $F_{vg_i}$ = fixed vine-growing cost farm $i$ | $v_{wm}$ = variable cost of the winery |

$F_{wm}$ = fixed cost of the winery |

**Table A1.** Impacts of floods over investments, production, revenues, vine-growing and wine-making costs, at individual ($\forall$ farm $i$) and system's level in a moment $t$

$$Imp_t = (I'_t - I_t) + (p + v_{vg} + v_{wm})\left( \sum_{i=1}^{n} q'_{i,t} - \sum_{i=1}^{n} q_{i,t} \right) \tag{A16}$$

That is, the impact of a flood at any moment $t$ comes given by the differences in investment and yield/production. It is worth to point out that, at farm level, the impact also comprises the redistributing effect driven by the individual share of the winery's fixed cots. To ensure the comparability of financial flows over time, discount factors are utilized.

**Appendix B: Disclosure of data sources and applications**

| | | Cooperative winery | | | | | | | Farm | | | | | | Plot | | World | | | |
|---|---|---|---|---|---|---|---|---|---|---|---|---|---|---|---|---|---|---|---|---|
| | | Size | Main Stages production | Cost structure | Financial data | Damage function | Sharing rules | Behavior when impact | Size | Tasks | Cost structure | Financial data | Damage function | Behavior when impact | Productivity | Damage function | Number of plots | Number of farms | In/out flood prone area proportions | Market price |
| **GIS** | EEA (2012) | | | | | | | | | | | | | | | | | | ✓ | |
| | IGN (2020) | | | | | | | | | | | | | | | | | ✓ | ✓ | |
| | MTES (2020) | | | | | | | | | | | | | | | | ✓ | ✓ | | |
| **Statistics** | FranceAgriMer (2012) | ✓ | | | | | | | | | | | | | ✓ | | | | | ✓ |
| | Agreste (2010) | | | | | | | | ✓ | | | | | | | | | | | |
| | FADN (2014) | | | | | | | | | | ✓ | ✓ | | | | | | | | |
| **Reports** | INSEE (2016) | | | | | | | | | | | | ✓ | | | | | | | |
| | CCMSA (2017) | | | | | | | | | | | | ✓ | | | | | | | |
| | Centre d'économie rurale (2017) | | ✓ | | | | | | | | | | | | | | | | | |
| | Centre d'économie rurale (2014) | | | | | | | | | | ✓ | | | | ✓ | | | | | |
| **Literature** | Chevet (2004) | | ✓ | | ✓ | | | | | | | | | | | | | | | |
| | Folwell and Castaldi (2004) | | ✓ | | | | | | | | | | | | | | | | | |
| | Battagliani et al. (2009) | | | | | | | | ✓ | | | | | | | | | | | |
| | Brémond (2011) | | | | | | | | | ✓ | | | | | | | | | | |
| | Biarnès and Touzard (2003) | | | | | | ✓ | | | | | | | | | | | | | |
| | Jarrige and Touzard (2001) | | | | | | ✓ | | | | | | | | | | | | | |
| | Touzard et al. (2001) | | | | | | ✓ | | | | | | | | | | | | | |
| **Interviews** | Experts | | | ✓ | ✓ | | | ✓ | | | ✓ | ✓ | | | ✓ | | | | | |
| | Winery CEOs | ✓ | ✓ | | | | ✓ | ✓ | | | | | | | | | ✓ | ✓ | | |
| | Vinegrowers | ✓ | | | | | | | | | | | | ✓ | | | | | | |

**Table B1.** Sources of information organized by element feature