# Peer review of "Are interactions important in estimating flood damage to economic entities?"

_Natural Hazards and Earth System Sciences, 2020_

## Referee Comment (RC1) · Anonymous Referee #1 · 21 Dec 2020

The manuscript addresses the important issue of flood indirect damage to complex productive system (CPS) al local level. The authors applied an agent-based model as a virtual laboratory for the ex-ante estimation of impacts over cooperative winemaking system (CWS). They investigated the interactions within and between the elements (plots, farm, cooperative) composing the CPS to quantify the links between them depending of the season. Despite the amount of work done and the data collected, the manuscript does not convey the substance of the work. It is too much fragmentated in subsections that it is almost impossible to read and understand by readers who are not expert of their calculation model. As it currently appears, the manuscript is more like a technical report than a scientific manuscript. The description of the model and the workflow are too synthetic, they need to be enlarged and supported by a workflow

diagram that could help the reader across the different simulations. In particular section 3, 4 and 5 should be completely restructured and better described. The figures should be better described in the text and it would be better to add the labels of the x-y axes and the unit of measurement, also specify which axis the various curves refer to. Please give an extended explanation of what the farm coping tactics are since they are important in the model results. The reviewer recommends the author improving the discussion, explaining their results in the light of past flooding events and making practical cases of the different interactions occurred between and within the damaged elements to better understand how the different damage (to buildings or to plots) had negatively impacted the CWS. It would also be important to understand how this ex ante analysis can help the economy of the wine sector.

---

## Referee Comment (RC2) · Anonymous Referee #2 · 16 Feb 2021

**Review Comments**

**Abstract :**

Suggest the abstract need to define direct and indirect , and possibly indicate the degree or estimation error that can be corrected for

**Manuscript .**

Interesting paper on a relevant topic.  It is clearly written but it is challenging to work through and I think it would benefit from some signposting that reminds the reader where (and why) we are up to in the argument.

**Some general points**

I think more could be made of the actual financial / costs  data obtained, and presented to show absolute costs : % changes can be difficulty to interpret : % of what? What do the uplift factors in the table of results actually mean to the baseline cost estimate used in a flood impact assessment or CBA ?

I also think the configurations could be grounded in what is observed  : what is the dominant case , and what are the main variations for the industry , perhaps with other configurations showing how estimates vary around a core /central estimate .

I think more explanation about the seasonal variation in the estimates , and importantly , the magnitude of the difference makes to the overall estimate (once seasonality and other issues are taken into account) relative to a 'careful' consideration of impacts on vine production and processing considered separately.

**More specific points**

| Introduction | Agree there is often confusion and an arbitrariness about the definition and classification of costs. Perhaps the paragraph could begin by making this point.  The use of the term 'flood damage' doesn't help either ; this implies a focus on damage to physical assets (stocks) and not to flows (incomes and expenditures).  It might be better to consider 'flood costs'.  It also point to the needs for a cost algorithm function to show what is in and what isn't  (see below)
Line 42 : so which definition are the authors using here? |
|---|---|
|  | The definitions are not independent of the purpose of the assessment : whether financial or economic, and whether concerned with costs: benefit or economic impact assessment. |
| 54 | What kind of values for example: the range in estimates of indirect (as defined here?) and direct can be considerable : 3 % to 30%  or more depending on impact sector , and guide on this |
| 54 | The use of static ratios or % of direct damage depends on the definition and estimate of direct costs in the first place: and this may vary? % of what? (see below) |

| 65 and para | Likely that ratio of direct and indirect will vary by impact sector /category , eg types of industry/ economic activity, transport, agriculture. As the authors know  In the agric case, damage to physical assets is relatively small : the biggest cost component is usually damage to crops- work in progress and evident in income loss and additional operating costs . (insurable asset losses are relatively small as a proportion.)     So how are we defining direct ? |
|---|---|
| 45 | perhaps should mention how this translates into GVA estimates and multipliers, with various assumptions about additionality/displacement |
| 105 | Suggest you say who the paper is aimed at |
| 115 | Is this costs to agriculture as a share of total event cost? |
| 125 | Perhaps clarify that flood costs here include asset damage as well as income/expenditure impacts (an important aspects of agricultural flooding) |
| | Perhaps make it clearer that these two impact categories, farm production and off-farm commodity processing would potentially be treated as separate impact categories in flood assessment.  This is said later  but emphasise more here, I think . |
| 170 onwards | Rather complicated to follow : rest on estimates of damage to assets plus impact on revenues and costs, including work in progress? |
| | Seems to largely rest on the assumptions  regarding the impact on the winery.  Estimates of flooding on the wine production areas can be based on ex 'farm gate ' effects .  The variation depends then mainly on the effects on the winery :so either the winery incurs 'direct' damage , because it is flooded or it indirect damage because, been though not flood,  the quality or quality of supply is affected : so what re the impacts on these two elements in the supply/value chain?  I think you are saying the underestimation is where the winery is safe from flooding, but takes a hit from not having grapes.  But if it does flood, the impacts are assessed as a loss of contents and process.  Hence why there is a big lift in your figures 4 and 4 .  You might make this (more) clear |
| | |
| 150 | Given actual cost data were collected it would be good to include absolute flood event cost estimates , and their make up/distribution between cost components |
| | A critical point is that that the quantitative results are given as a % of baseline: but what are the bae line costs.  The use of coefficients and weights to assess 'indirect'  costs depend heavily on what the baseline estimate is > And assume that the baseline here is the sum of the two impact categories considered separately.  I note that the estimates are by flood extent, but what are the costs per ha of vine flooded    , or per unit capacity of wintery ? |
| 250 | Figure 2: what's the top dotted blue line |
| 300 and thereabout | The assumptions and configurations are challenging to follow,  How representative are these configurations of what is observed in practice: is the size exposure configuration that gives the highest cost increase common ? or has the industry already adapted to flood risk? |
| 300 | It would be useful to produce a cost function that summarises the type of costs , even better it would be good to produce estimates of costs showing the make up of the cost estimates for the different scenarios /configurations. There appears to be 'damage' to asset 'stocks'  as well as to income/expenditure flows: what's the proportions of these. Not quite sure what is meant by material damage . Is there an underlying flood evet cost algorithm? |

| | |
|---|---|
| | |
| | 'Concrete' flood, suggest rephrase |
| 360 and onwards | Would be good to have some cost estimates , as suggested above , and this would help show the scale of the differences in the estimates with an without the connections |
| 420 onwards | The results suggest, as far as I can see, that the main differences (either in costs by configuration or in costs relative to the baseline ) are due to autumn and winter flooding. What is the underlying seasonal distribution of flood costs ?

More explanation of what to look for in the figures would be good, especially on observed variation (or lack of it) |
| 490 | I think some of the points in the conclusions , night better go to reinforce the discussions : perhaps there should be a section on discussion of results and what they mean , in their and particularly, in practice, linked to the point s made in the introductory sections. |
| 495 onwards | It seems as though the cost estimates depend on seasonality as it determines where the grapes are in ex--vine storage and processing system, so the assessment of costs (relative to the baseline) largely depends on damage to stocks and flows of grapes in the system, which is seasonally defined.   So I am asking why would not a seasonally based estimate of damage accommodate this for the production (on the farm) and for the vinery, reflecting the dominant configuration .   (A coping strategy might also be to important grapes from elsewhere to keep the process going, at a cost) ) |
| | I note the points made about overestimate and underestimation of 'indirect' : hence the importance of defining indirect |
| | Conclusions :e a lot of this is discussion and could beneficially put in a section called that

You say the approach is too costly :could  estimates be built into the cost algorithm for representative configurations of the industry to allow for these so-called 'indirect' impacts |

---

## Author Comment (AC1) · 22 Mar 2021

We thank you for your feedback and your suggestions for improving our manuscript. Below we provide our replies to the general comments of this referee, using the following structure: the referee's comment is highlighted in bold font, whereas the answer of the authors is included in normal font.

- **It (the article) is too much fragmentated in subsections that it is almost impossible to read and understand by readers who are not expert of their calculation model. As it currently appears, the manuscript is more like a technical report than a scientific manuscript.**

Answer of the authors:

- We thank referee #1 for this observation, but we partly disagree with it, the global structuration of our manuscript is organized as the one of an article and not of a technical report. Its structure has been carefully considered to provide the reader the key elements to understand the interest of our work, our method, the protocol of simulations followed to test the hypotheses enounced, the results obtained and the meaning of such results.

- Concerning the use of subsections, we will reconsider carefully their use in the next version, and limit our structuration in tow levels (section and subsection).

- Regarding the expertise required on our model, we provide a full answer in the next point.

- **The description of the model and the workflow are too synthetic, they need to be enlarged and supported by a workflow.**

  Answer of the authors:

  - We thank referee #1 for this suggestion.

  - The current description provided summarizes the key points that the reader needs to understand the work accomplished in our article. For those readers who would like to deepen their knowledge in our model, the authors have also provided the reference to Nortes Martinez, D., Grelot, F., and Rouchier, J.: COOPER - Flood impacts over Cooperative Winemaking Systems, CoMSES Computational Model Library, https://www.comses.net/codebases/6038/releases/1.0.1/, version 1.0.1, 2019. This is the repository of the COOPER model in the COMSES model library, where readers have free access to a 76-page technical report

documenting the model –according the precepts of the ODD protocol– as well as to the source code of the model.

– Nonetheless, the referee's suggestion is duly noted and a more extended description of the model will be provided, including workflow diagrams, a better characterization of the coping tactics (also suggested by the referee #1. See below), impacts calculations and cost distributions.

• **[. . . ] section 3, 4 and 5 should be completely restructured and better described.**

Answer of the authors:

– We thank referee #1 for this suggestion.
– Concerning section 3, we propose to make a new presentation based on what we explained in the previous point.
– Concerning section 4, we propose a restructuration, limiting the number of subsections.
– Concerning section 5, we do not understand what should be restructured as in this section the use of subsections is limited.

• **The figures should be better described in the text and it would be better to add the labels of the x-y axes and the unit of measurement, also specify which axis the various curves refer to.**

Answer of the authors:

– The authors thank the referee #1 for this observation.
– The description of the figures will be better described, labels to x-y axis will be provided next to the axis in addition to the labels already provided in the remarks that accompany each figure, and secondary axis will be suppressed to avoid confusions.

- **Please give an extended explanation of what the farm coping tactics are since they are important in the model results.**

  Answer of the authors:

  – We thanks referee #1 for this suggestion.
  – The authors agree with the comments of the referee #1. This remark will be included in the extended version of section 3.

- **The referee recommends the author improving the discussion, explaining their results in the light of past flooding events and making practical cases of the different interactions occurred between and within the damaged elements to better understand how the different damage (to buildings or to plots) had negatively impacted the CWS.**

  Answer of the authors:

  – The authors appreciate the recommendation of the referee #1 to strengthen the discussion of the paper and agree on the convenience of such an inclusion.
  – Unfortunately, available information on past flood events is, at best, fragmented and rarely available in public form with the level of detail that this comparison would need. In particular, very few farms are insured and the sinistrality data are unusable for validation or discussion. Indeed, to overcome these problems in the development of the COOPER model, authors have targeted past events and used different elicitation methods available to them: GIS, field interviews, professional expertise, census and statistical data as well as analysis of technical reports and scientific literature. The massive amount of data collected allowed the authors to reconstruct the way a real French CWS works and reacts to a flood event. Then, over this

meta-model, authors were able to build the COOPER model for flood impact simulation.

– This approach not only enables us to overcome the challenges provoked by the data quality and availability in the model design but also to allows us to validate our model. Indeed, insofar as the data to validate the model COOPER is limited, we opted for the so-called conceptual validation (Rykiel Jr., 1996). This kind of validation relies on the theoretical plausibility, accuracy and justifiability of the relations cause-effect built in the model.

– Consequently, although the authors agree with the referee's recommendation, such an comparison exercise is not possible with the data publicly available.

– The authors will include this argument in the next version

• **It would also be important to understand how this ex ante analysis can help the economy of the wine sector.**

Answer of the authors:

– The authors appreciate the recommendation of the referee #1 to strengthen the discussion of the paper, we will mention this point in our next version.

– Nonetheless, the goal of the article is not to study the wine sector per se but rather, using a specific case study taken from the wine sector, to illustrate how important could be to consider interactions for flood damage assessment at microeconomic, local levels. From this point of view, we discuss the relevance of our work for the analysis of any local organization or supply chain that works in a similar way as the cooperative winemaking system. This is the reason not to focus specifically the on wine sector in our discussion and conclusion. Indeed, as stated in Meyer et al., 2013, "most stakeholders are interested in assessing the indirect impact of various types

of events, both large and small, at micro (cities) or meso (catchment) scale, with or without risk mitigation measures. However, most of the methods discussed can only assess the impacts of an extreme event on the national or sometimes regional scale". Our work benefits from the potential of Agent-based models to become computational laboratories in which to evaluate the response of a system to an external perturbation of variable magnitude. Indeed, the goal of the COOPER model is to evaluate the potential disruption caused by different floods over a system bounded to a local, very concrete, highly detailed, monosectoral and spatially-explicit supply chain. Furthermore, impacts and disruptions in the COOPER model can be studied either at the level of one individual or groups of individuals. Therefore the model has a very clear microeconomic orientation. The goal of the present article is to present how current practices in damage assessment at local level can benefit from more thorough and spatially-explicit representations of local networks.

V. Meyer, N. Becker, V. Markantonis, R. Schwarze, J. C. J. M. van den Bergh, L. M. Bouwer, P. Bubeck, P. Ciavola, E. Genovese, C. Green, S. Hallegatte, H. Kreibich, Q. Lequeux, I. Logar, E. Papyrakis, C. Pfurtscheller, J. Poussin, V. Przyluski, A. H. Thieken, and C. Viavattene. Review article: Assessing the costs of natural hazards - state of the art and knowledge gaps. Natural Hazards and Earth System Sciences, 13(5):1351–1373, 2013.

E. Rykiel Jr., Testing ecological models: the meaning of validation, Ecological Modelling 90 (1996) 229 – 244. doi:http://dx.doi.org/10.1016/0304-3800(95)00152-2.

---

## Author Comment (AC2) · 22 Mar 2021

We thank you for your feedback and your insights to improve the quality of our manuscript. Below we provide our replies to both the general comments and to the specific points made by the referee, using the following structure: the referee's comment is highlighted in bold font, whereas the answer of the authors is included in normal font.

**Referee's general comments**

- **Interesting paper on a relevant topic. It is clearly written but it is challenging to work through and I think it would benefit from some signposting that**

**reminds the reader where (and why) we are up to in the argument.**

Answer of the authors:

– Thank you very much for the overall assessment. The recommendation is duly noted by the authors.

• **I think more could be made of the actual financial/costs data obtained, and presented to show absolute costs: % changes can be difficulty to interpret: % of what? What do the uplift factors in the table of results actually mean to the baseline cost estimate used in a flood impact assessment or CBA ?**

Answer of the authors:

– The authors agree with the statement of the referee. Indeed, the COOPER model enables the analyst to perform financial analysis on cash flows in the system throughout the simulation. Such analysis though are out of the scope of this work. Here we assess the effect that taking into account explicit interactions between entities in a local productive chain may have on the assessment of flood damages in comparison to the more standard practice of not considering them.

– To assess such an effect, the authors consider that percentages (%) of variation compared to a baseline are more generic (less case-oriented), illustrative and easily comparable than absolute values of monetary assessments. Thus they transmit the message more directly and efficiently.

– These percentages represent the variation of the monetary value of the flood impact in the cooperative winemaking system of a given modality of interaction or configuration with respect to the baseline for the period simulated. The baseline is fixed as the *no interaction* modality as it represents the standard in nowadays professional practice. Namely, to take into account the entities in the system but not the interactions among them.

– To calculate the absolute impact, the COOPER model compares two scenarios: the business as usual scenario (simulation of n years without floods) and the simulated flood scenario (simulation of n years with a flood), as it is described in section 3.6.9

• **I also think the configurations could be grounded in what is observed: what is the dominant case, and what are the main variations for the industry, perhaps with other configurations showing how estimates vary around a core /central estimate.**

Answer of the authors:

– The authors agree with the observation of the referee #2. However the information that could conduct us to be able to establish a dominant case of network configuration (hence exposure profiles for farmers) in the cooperative winemaking system are neither publicly available nor willingly divulged by the stakeholders.

– At the same time the exercise suggested by the referee is a version of what we do in our article: setting the flood damage configuration of links named no interaction as a baseline (current practice in CBA) to calculate, then, the difference in the monetary value of the flood damage between the baseline and a given scenario.

– The authors will make this more explicit in the next version

• **I think more explanation about the seasonal variation in the estimates, and importantly, the magnitude of the difference makes to the overall estimate (once seasonality and other issues are taken into account) relative to a 'careful' consideration of impacts on vine production and processing considered separately.**

Answer of the authors:

– The authors thanks the referee for this proposition. One of the challenges was, indeed, to take into account the seasonality of the processes of the different entities of the model. All the processes modeled and presented in section 3 take seasonality into account and are nested. The interest of the model used is to allow the analysis of the effects of the simultaneous consideration of seasonality for the different entities in comparison with scenarios where these interactions are not considered. The authors will consider this remark in the next version by making the entanglement of seasonal processes more explicit in section 3.

**Referee's specific points**

- **Agree there is often confusion and an arbitrariness about the definition and classification of costs. Perhaps the paragraph could begin by making this point. The use of the term 'flood damage' doesn't help either; this implies a focus on damage to physical assets (stocks) and not to flows (incomes and expenditures). It might be better to consider 'flood costs'. It also point to the needs for a cost algorithm function to show what is in and what isn't (see below).**

**Line 42: so which definition are the authors using here?**

**The definitions are not independent of the purpose of the assessment: whether financial or economic, and whether concerned with costs: benefit or economic impact assessment.**

Answer of the authors:

- The authors appreciate the terminology suggestion since the debates on the best terminology have been present from day one.

- The authors would like to point out that the goal of this paragraph is not to inform the reader of the definitions used in the paper, but to show the diversity of definitions of indirect damage that coexist in the literature. Inasmuch as the COOPER model is spatialized and dynamic, its design grants each potential user the flexibility necessary to fit his/her definitions of indirect impacts, as the ones given by, e.g. Cochrane, 2004; Meyer et al., 2013; or Penning-Rowsell or Greene, 2000.

In other words, the COOPER model simulates an encapsulated system (the CWS) where each entity performs different roles, creating a flow of inputs and outputs between them. When a flood hits the system both performance and flows can be compromised. By comparing a simulation with no floods with a simulation with a flood we can determine the impact that the flood has had on the system, regardless any classification of damages/impacts. Given that the study that we are presenting is done at a system level, we do not need to classify specific damages in direct and indirect. That being said, if we were to descend to the level of the entity, e.g. the farm, we agree with the referee #2 that we would need to ascribe to one specific classification to designate which damage is direct or indirect from the point of view of the farm. The same applies if we were studying the impact at the system level looking at interactions with other systems. Therefore, taking into account that we study the impacts of a flood within the boundaries of our system, and that we do it at a system level because the goal of the paper does not demand damage disaggregation, the authors do not consider pertinent to ascribe to one or another set of definitions. - The authors propose to make this point more explicit in the next version of the paper.

- **What kind of values for example: the range in estimates of indirect (as defined here?) and direct can be considerable: 3 % to 30% or more depending on impact sector, and guide on this**

**The use of static ratios or % of direct damage depends on the definition and**

**estimate of direct costs in the first place: and this may vary? % of what?**

**Likely that ratio of direct and indirect will vary by impact sector /category, eg types of industry/ economic activity, transport, agriculture. As the authors know In the agric case, damage to physical assets is relatively small: the biggest cost component is usually damage to crops- work in progress and evident in income loss and additional operating costs. (insurable asset losses are relatively small as a proportion.) So how are we defining direct ?**

**[. . . ] perhaps should mention how this translates into GVA estimates and multipliers, with various assumptions about additionality/displacement**

Answer of the authors:

- The authors would like to point out that the statements regarding computational general equilibrium models and input-output models (CGE/I-O hereafter) do not pursue to discuss in-depth CGE/I-O, insofar as these models are out of the scope of the work presented in the article. The article's scope is microeconomic, monosectoral and local as opposed to macroeconomic, multisectoral and regional/national. As noted by Green et al., 2011, and Meyer et al., 2013, CGE/I-O have been indeed successfully implemented at national and/or regional levels, though their potential to provide useful information to decision-makers when the economic disruptions of floods might vanish before reaching the aforementioned levels is debatable. Agent-based models (ABMs hereafter) have the potential to fill the gap left by CGE/I-O, and become useful tools to evaluate flood impacts in local communities, thoroughly representing the complexities faced at this local level. In that sense, ABMs represent complementary tools to CGE/I-O in the flood damage assessment.

- The COOPER model does not use fixed coefficients of direct damage to assess the indirect damage. Rather it rests over a vector of four key variables: i) production; ii) revenues; iii) costs; and iv) investments and reinvestments. This last variable serves us to group all reparations to be done in the system after a flood, reinvestments in plants and materials and planned investments independent of the flood. The impacts are then calculated by differences between two kinds of scenarios, as explained above: the business as usual scenario (simulation of n years without floods) and the simulated flood scenario (simulation of n years with a flood), as it is described in section 3.6.9.

- Consequently, the authors considered superfluos (and potentially misguiding) a review of literature that goes too deep into the details of the CGE/I-O models.

- That being said, we understand this and some other remarks made by the referee #2 that aim at improving the model description for the reader's coomprehension sake. Hence the authors will provide an extended description of the model, including workflow diagrams, a better characterization of the coping tactics (also suggested by the referee #1), impacts calculations and cost distributions.

- **Suggest you say who the paper is aimed at.**

  Answer of the authors:

  – The authors appreciate the suggestion of the referee #2. The paper is aimed at the whole community of researchers in economic impacts of natural catastrophes and economics of natural disasters.

- **Is this costs to agriculture as a share of total event cost?**

  Answer of the authors:

  – The interpretation of the referee #2 is correct. We will propose a better formulation to make it easier to understand.

- **Perhaps clarify that flood costs here include asset damage as well as income/expenditure impacts (an important aspects of agricultural flooding) Perhaps make it clearer that these two impact categories, farm production and off-farm commodity processing would potentially be treated as separate impact categories in flood assessment. This is said later but emphasize more here, I think.**

  Answer of the authors:

  – The authors appreciate the suggestion of the referee #2. We will propose a better formulation to make it easier to understand.

- **Given actual cost data were collected it would be good to include absolute flood event cost estimates, and their make up/distribution between cost components.**

**A critical point is that that the quantitative results are given as a % of baseline: but what are the base line costs. The use of coefficients and weights to assess 'indirect' costs depend heavily on what the baseline estimate is > And assume that the baseline here is the sum of the two impact categories considered separately. I note that the estimates are by flood extent, but what are the costs per ha of vine flooded, or per unit capacity of wintery ?**

**Would be good to have some cost estimates, as suggested above, and this would help show the scale of the differences in the estimates with an without the connections**

Answer of the authors:

- The authors thank the suggestion of the referee #2. The COOPER model, developed by the authors, does not use coefficients to measure indirect impacts.

Rather, it compares the values of four key variables as explained above. We chose not to present every detail in the current version of this article as there are accessible in COMSES. However, as this seems important to better understand our work, we propose, on the one hand, to address this point in an extended description of the COOPER model, and, on the other hand, to add a little section to give absolute cost estimates in the baseline scenarios.

- **Rather complicated to follow: rest on estimates of damage to assets plus impact on revenues and costs, including work in progress?**

  Answer of the authors:

  – The interpretation of referee #2 is correct. We will propose a better formulation to make it easier to understand.

- **Seems to largely rest on the assumptions regarding the impact on the winery. Estimates of flooding on the wine production areas can be based on ex 'farm gate' effects. The variation depends then mainly on the effects on the winery: so either the winery incurs 'direct' damage, because it is flooded or it indirect damage because, been though not flood, the quality or quality of supply is affected: so what are the impacts on these two elements in the supply/value chain? I think you are saying the underestimation is where the winery is safe from flooding, but takes a hit from not having grapes. But if it does flood, the impacts are assessed as a loss of contents and process. Hence why there is a big lift in your figures 4 and 4. You might make this (more) clear.**

  Answer of the authors:

  – The recommendation of the referee #2 is duly noted. The new description of the model COOPER will address the point of direct/indirect impacts so the results obtained are perfectly clear for the reader

- **Figure 2: what's the top dotted blue line.**

  Answer of the authors:

  – The accumulated number of plots at a given position. We will propose a better formulation to make it easier to understand.

- **The assumptions and configurations are challenging to follow, How representative are these configurations of what is observed in practice: is the size exposure configuration that gives the highest cost increase common ? or has the industry already adapted to flood risk?**

  Answer of the authors:

  – The section 4 is going to be restructured to restrain the number of sub-subsections. We hope that this change will make the section flow better.

  – The information that could conduct us to be able to establish a representative configuration for the cooperative winemaking system are neither publicly available nor willingly divulged by the stakeholders. Thus we cannot establish a "standard" in relation to configurations and spatial locations.

  – Furthermore, it is likely that "real life" configurations are very different from each other and sometimes overexposed small to mid size farms whereas other times big farms stocked riverine lands. That is why we think that the exercise done has interest: between the extremes, we have all potential "real life" situations. Nonetheless, the work we are presenting here is not an analysis of a particular system, during a particular flood. Rather, it is a plausible system based in "real world" systems facing a multitude of different floods, that shows us the misestimations in which we incur when we choose to ignore the links among entities in a local productive chain.

  – These misestimations are all the more important when flood impacts assessments are utilized to design and calculate the mechanisms of monetary

compensations to be implemented in rural communities that are used to protect urban and industrial areas from floods.

- **It would be useful to produce a cost function that summarizes the type of costs, even better it would be good to produce estimates of costs showing the make up of the cost estimates for the different scenarios /configurations. There appears to be 'damage' to asset 'stocks' as well as to income/expenditure flows: what's the proportions of these. Not quite sure what is meant by material damage. Is there an underlying flood evet cost algorithm?.**

  Answer of the authors:

  – The recommendation of the referee #2 is duly noted. The new description of the model COOPER will address this point.

- **Concrete' flood, suggest rephrase.**

  Answer of the authors:

  – The authors agree and they will rephrase it to "given flood".

- **The results suggest, as far as I can see, that the main differences (either in costs by configuration or in costs relative to the baseline ) are due to autumn and winter flooding. What is the underlying seasonal distribution of flood costs ? More explanation of what to look for in the figures would be good, especially on observed variation (or lack of it)**

  Answer of the authors:

  – The recommendation of the referee #2 is duly noted. The new description of the model COOPER will address this point

- **I think some of the points in the conclusions, might better go to reinforce the discussions: perhaps there should be a section on discussion of results and what they mean, in their and particularly, in practice, linked to the points made in the introductory sections.**

  Answer of the authors:

  – The authors thank the referee #2 for the remark. Following the suggestion of the referee #2, the new version of the article would split the contents of this section into *discussion* and *conclusion*

- **It seems as though the cost estimates depend on seasonality as it determines where the grapes are in ex–vine storage and processing system, so the assessment of costs (relative to the baseline) largely depends on damage to stocks and flows of grapes in the system, which is seasonally defined. So I am asking why would not a seasonally based estimate of damage accommodate this for the production (on the farm) and for the vinery, reflecting the dominant configuration. (A coping strategy might also be to important grapes from elsewhere to keep the process going, at a cost)**

  Answer of the authors:

  – The authors thank the referee #2 for the remark and the suggestion. As we understand the statement, we would like to point out that our work goes one step further of what the referee is proposing. Sure, we could make the assumption that stocks are either in one place or another -even using a probabilistic distribution behind based on a supposed dominant way of working- and do not take into account the flow of goods from one entity to another according to their schedules. This is somehow the exercise proposed in the case of *interactions within activities*: since information regarding losses of harvest does not flow from farm to winery, in those cases in which a flood

impacts the winery, an estimation of the loss should be done and that might
lead to problems of double accountability of damages.

– With the COOPER model we go one step further and we can know exactly
where the stock is so estimations do not need to take place.

– Concerning the suggestion on the coping tactic, we agree that, in absolute
terms, it would be an effective one. However the plausibility of such a tac-
tic is inevitably linked to the reality of the CWS. As it is today, cooperative
winemaking systems group farmers that decide to share the property of pro-
ductive means to process their harvest and create their products employing
professional experts for the day-to-day management and commercialization.
The possibility that those farmers willingly accept to buy grapes from other
producers is unlikely. Furthermore, if those farmers belong to a so-called
*Appellation d'origine contrôlée*, they are subject to very strict rules regard-
ing the quality and origin of the product (even the origin of the soil), if they
want to keep the label.

– These institutional constraints made us decide to focus on the two more
plausible coping tactics evoked by the interviewed farmers. We will add this
point to our discussion.

• **You say the approach is too costly: could estimates be built into the cost al-
gorithm for representative configurations of the industry to allow for these
so-called 'indirect' impacts.**

Answer of the authors:

– The idea we wanted to pass to the readers with such a statement is that the
approach is too costly if it were to be implemented with the same amount
of detail at a different level other than the local one, given the thoroughness
with which the system is described and "translated" into a model. The best

approach we can foresee is to analyze in depth different local communities using detailed models, such as the COOPER, that feed regional or national models in some sort of nested structure.

H. C. Cochrane. Indirect losses from natural disasters: Measurement and myth. In Y. Okuyama and S. E. Chang, editors, Modeling Spatial and Economic Impacts of Disasters, Advances in Spatial Science, chapter 3, pages 37–52. Springer Berlin Heidelberg, 2004. doi:10.1007/978-3-540-24787-6_3.

C. Green, C. Viavattene, and P. Thompson. Guidance for assessing flood losses. CONHAZ WP6 Final Report, Flood Hazard Research Center - Middlesex University, Sept. 2011.

V. Meyer, N. Becker, V. Markantonis, R. Schwarze, J. C. J. M. van den Bergh, L. M. Bouwer, P. Bubeck, P. Ciavola, E. Genovese, C. Green, S. Hallegatte, H. Kreibich, Q. Lequeux, I. Logar, E. Papyrakis, C. Pfurtscheller, J. Poussin, V. Przyluski, A. H. Thieken, and C. Viavattene. Review article: Assessing the costs of natural hazards - state of the art and knowledge gaps. Natural Hazards and Earth System Sciences, 13(5):1351–1373, 2013.

E. C. Penning-Rowsell and C. H. Green. New insights into the appraisal of flood-alleviation benefits: (1) flood damage and flood loss information. Water and Environment Journal, 14(5):347–353, 2000. doi:10.1111/j.1747-6593.2000.tb00272.x.

---

## Author Response (AR1)

**Summary of changes and comments regarding observations and recommendations from referee #1**

We thank the referee #1 for his feedback and suggestions for improving our manuscript. Below we provide our replies to the general comments of this referee, using the following structure: the referee's comment is highlighted in bold font, whereas the answer of the authors is included in normal font.

- **It (the article) is too much fragmentated in subsections that it is almost impossible to read and understand by readers who are not expert of their calculation model. As it currently appears, the manuscript is more like a technical report than a scientific manuscript.**

Answer of the authors:

- The whole article has been restructured to accommodate a more standard structure in scientific communications.

- A new, more detailed description of the model has been provided.

- **The description of the model and the workflow are too synthetic, they need to be enlarged and supported by a workflow.**

Answer of the authors:

- The model description is now part of a larger section "Method". This section starts with an overview in which we include a new workflow for our paper.

- The model description has been rewritten to offer a more insightful description using a less fragmented structure. A new model workflow has been included as well as a new section on the calculation of impacts. This last section is completed with a mathematical annex at the end of the article

- **[. . . ] section 3, 4 and 5 should be completely restructured and better described.**

Answer of the authors:

- Sections 3 and 4 have been integrated in the new section "Method". Both of them have been restructured.

- Section 5 has been included in a larger section "Results" along with section 6 and a new section "Baseline". After careful consideration, we have decided to keep the structure of the section 5 as it was since we do not understand what should be restructured.

- **The figures should be better described in the text and it would be better to add the labels of the x-y axes and the unit of**

**measurement, also specify which axis the various curves refer to.**

Answer of the authors:

- Figures were modified following the recommendation of the referee #1. However the result made the resulting figure more confusing and busy. Thus, we opted for keeping them as they were, providing the information on axes, units, etc at the bottom of each figure.

- **Please give an extended explanation of what the farm coping tactics are since they are important in the model results.**

Answer of the authors:

- The new model description includes a more detailed explanation of the coping tactics and its effects.

- **The referee recommends the author improving the discussion, explaining their results in the light of past flooding events and making practical cases of the different interactions occurred between and within the damaged elements to better understand how the different damage (to buildings or to plots) had negatively impacted the CWS.**

Answer of the authors:

- The new version of the article includes a new section "Discussion" and a new section "Conclusion". Unfortunately, available information on past flood events is, at best, fragmented and rarely available in public form with the level of detail that this comparison would need. In particular, very few farms are insured and the sinistrality data are unusable for discussion. Consequently, although the authors agree with the referee's recommendation, such an comparison exercise is not possible with the data publicly available and we have not been able to include it.

- The authors have targeted past events and used different elicitation methods available to them (GIS, field interviews, professional expertise, census and statistical data as well as analysis of technical reports and scientific literature) to overcome those problems in the development of the COOPER model. The amount of data collected allowed the authors to reconstruct the way a real French CWS works and reacts to a flood event. Then, over this meta-model, authors were able to build the COOPER model for flood impact simulation. This approach enables us to overcome the challenges provoked by the data quality and availability in the model design but also to allows us to validate our model. Indeed, insofar as the data to validate the model COOPER is limited, we opted for the so-called conceptual validation (Rykiel Jr., 1996). This kind of validation relies on the theoretical plausibility, accuracy and justifiability of the relations cause-effect built in the model.

- **It would also be important to understand how this ex ante analysis can help the economy of the wine sector.**

Answer of the authors:

- The authors thank the referee #1 for the recommendation. Nonetheless, the goal of the article is not to study the wine sector per se but rather, using a specific case study taken from the wine sector, to illustrate how important could be to consider interactions for flood damage assessment at microeconomic, local levels. From this point of view, we discuss the relevance of our work for the analysis of any local organization or supply chain that works in a similar way as the cooperative winemaking system. For this reason, we have provided an statement on how this kind of ex-ante analysis can help local communities and concerning actors and stakeholders, but we consider best not to focus specifically on wine sector. Indeed, as stated in Meyer et al., 2013, "most stakeholders are interested in assessing the indirect impact of various types of events, both large and small, at micro (cities) or meso (catchment) scale, with or without risk mitigation measures. However, most of the methods discussed can only assess the impacts of an extreme event on the national or sometimes regional scale". Our work benefits from the potential of Agent-based models to become computational laboratories in which to evaluate the response of a system to an external perturbation of variable magnitude. Indeed, the goal of the COOPER model is to evaluate the potential disruption caused by different floods over a system bounded to a local, very specific, highly detailed, monosectoral and spatially-explicit supply chain. Furthermore, impacts and disruptions in the COOPER model can be studied either at the level of one individual or groups of individuals. Therefore the model has a very clear microeconomic orientation. The goal of the present article is to present how current practices in damage assessment at local level can benefit from more thorough and spatially-explicit representations of local networks.

V. Meyer, N. Becker, V. Markantonis, R. Schwarze, J. C. J. M. van den Bergh, L. M. Bouwer, P. Bubeck, P. Ciavola, E. Genovese, C. Green, S. Hallegatte, H. Kreibich, Q. Lequeux, I. Logar, E. Papyrakis, C. Pfurtscheller, J. Poussin, V. Przyluski, A. H. Thieken, and C. Viavattene. Review article: Assessing the costs of natural hazards - state of the art and knowledge gaps. Natural Hazards and Earth System Sciences, 13(5):1351–1373, 2013.

E. Rykiel Jr., Testing ecological models: the meaning of validation, Ecological Modelling 90 (1996) 229 – 244. doi:http://dx.doi.org/10.1016/0304-3800(95)00152-2.

**Summary of changes and comments regarding observations and recommendations from referee #2**

We thank the referee #2 for his feedback and insights to improve the quality of our manuscript. Below we provide our replies to both the general comments and to the specific points made by the referee, using the following structure: the referee's comment is highlighted in bold font, whereas the answer of the authors is included in normal font.

**Referee's general comments**

- **Interesting paper on a relevant topic. It is clearly written but it is challenging to work through and I think it would benefit from some signposting that reminds the reader where (and why) we are up to in the argument.**

Answer of the authors:

- We article has been restructured and new content has been added to improve the readability of the paper.

- **I think more could be made of the actual financial/costs data obtained, and presented to show absolute costs: % changes can be difficulty to interpret: % of what? What do the uplift factors in the table of results actually mean to the baseline cost estimate used in a flood impact assessment or CBA ?**

Answer of the authors:

- The new version of the article includes a new section that presents the estimation of impacts for the baseline in absolute terms.

- Nonetheless, for the visualization of the differences between this baseline and the rest of the simulations proposed in our experiments, the authors have kept the representation in terms of percentage (%). We consider that percentages (%) of variation compared to a baseline are more generic (less case-oriented), illustrative and easily comparable than absolute values of monetary assessments. Thus they transmit the message more directly and efficiently.

- These percentages represent the variation of the monetary value of the flood impact in the cooperative winemaking system of a given modality of interaction or configuration with respect to the baseline for the period simulated.

- **I also think the configurations could be grounded in what is observed: what is the dominant case, and what are the main variations for the industry, perhaps with other configurations showing how estimates vary around a core /central estimate.**

Answer of the authors:

- The authors agree with the observation of the referee #2. However the information that could conduct us to be able to establish a dominant case of network configuration (hence exposure profiles for farmers) in the cooperative winemaking system are neither publicly available nor willingly divulged by the stakeholders.

- At the same time the exercise suggested by the referee is a version of what we do in our article: setting the flood damage configuration of links named no interaction as a baseline (current practice in CBA) to calculate, then, the difference in the monetary value of the flood damage between the baseline and a given scenario.

- **I think more explanation about the seasonal variation in the estimates, and importantly, the magnitude of the difference makes to the overall estimate (once seasonality and other issues are taken into account) relative to a 'careful' consideration of impacts on vine production and processing considered separately.**

Answer of the authors:

- The authors thanks the referee for this proposition. One of the challenges was, indeed, to take into account the seasonality of the processes of the different entities of the model. All the processes modeled and presented in section 3 take seasonality into account and are nested. The interest of the model used is to allow the analysis of the effects of the simultaneous consideration of seasonality for the different entities in comparison with scenarios where these interactions are not considered.
- The authors have provided a more explicit description of the entanglement of seasonal processes in the new model description.

**Referee's specific points**

- **Agree there is often confusion and an arbitrariness about the definition and classification of costs. Perhaps the paragraph could begin by making this point. The use of the term 'flood damage' doesn't help either; this implies a focus on damage to physical assets (stocks) and not to flows (incomes and expenditures). It might be better to consider 'flood costs'. It also point to the needs for a cost algorithm function to show what is in and what isn't (see below).**

**Line 42: so which definition are the authors using here?**

**The definitions are not independent of the purpose of the assessment: whether financial or economic, and whether concerned with costs: benefit or economic impact assessment.**

Answer of the authors:

- The authors appreciate the terminology suggestion since the debates on the best terminology have been present from day one.
- The authors would like to point out that the goal of this paragraph is not to inform the reader of the definitions used in the paper, but to show the diversity of definitions of indirect damage that coexist in the literature. Inasmuch as the COOPER model is spatialized and dynamic, its design grants each potential user the flexibility necessary to fit his/her definitions of indirect impacts, as the ones given by, e.g. Cochrane, 2004; Meyer et al., 2013; or Penning-Rowsell or Greene, 2000.

In other words, the COOPER model simulates an encapsulated system (the CWS) where each entity performs different roles, creating a flow of inputs and outputs between them. When a flood hits the system both performance and flows can be compromised. By comparing a simulation with no floods with a simulation with a flood we can determine the impact that the flood has had on the system, regardless any classification of damages/impacts. Given that the study that we are presenting is done at a system level, we do not need to classify specific damages in direct and indirect. That being said, if we were to descend to the level of the entity, e.g. the farm, we agree with the referee #2 that we would need to ascribe to one specific classification to designate which damage is direct or indirect from the point of view of the farm. The same applies if we were studying the impact at the system level looking at interactions with other systems. Therefore, taking into account that we study the impacts of a flood within the boundaries of our system, and that we do it at a system level because the goal of the paper does not demand damage disaggregation, the authors do not consider pertinent to ascribe to one or another set of definitions. - The authors have made this point more explicit in the new model description (current section 3.2.2).

- **What kind of values for example: the range in estimates of indirect (as defined here?) and direct can be considerable: 3 % to 30% or more depending on impact sector, and guide on this**

**The use of static ratios or % of direct damage depends on the definition and estimate of direct costs in the first place: and this may vary? % of what?**

**Likely that ratio of direct and indirect will vary by impact sector /category, eg types of industry/ economic activity, transport, agriculture. As the authors know In the agric case, damage to physical assets is relatively small: the biggest cost component is usually damage to crops-work in progress and evident in income loss and additional operating costs. (insurable asset losses are relatively small as a proportion.) So how are we defining direct ?**

**[. . . ] perhaps should mention how this translates into GVA estimates and multipliers, with various assumptions about additionality/displacement**

Answer of the authors:

- The authors would like to point out that the statements regarding computational general equilibrium models and input-output models (CGE/I-O hereafter) do not pursue to discuss in-depth CGE/I-O, insofar as these models are out of the scope of the work presented in the article. The article's scope is microeconomic, monosectoral and local as opposed to macroeconomic, multisectoral and regional/national. As noted by Green et al., 2011, and Meyer et al., 2013, CGE/I-O have been indeed successfully implemented at national and/or regional levels, though their potential to provide useful information to decision-makers when the economic disruptions of floods might vanish before reaching the aforementioned levels is debatable. Agent-based models (ABMs hereafter) have the potential to fill the gap left by CGE/I-O, and become useful tools to evaluate flood impacts in local communities, thoroughly representing the complexities faced at this local level. In that sense, ABMs represent complementary tools to CGE/I-O in the flood damage assessment.

- The COOPER model does not use fixed coefficients of direct damage to assess the indirect damage. Rather it rests over a vector of four key variables: i) production; ii) revenues; iii) costs; and iv) investments and reinvestments. This last variable serves us to group all reparations to be done in the system after a flood, reinvestments in plants and materials and planned investments independent of the flood. The impacts are then calculated by differences between two kinds of scenarios, as explained above: the business as usual scenario (simulation of n years without floods) and the simulated flood scenario (simulation of n years with a flood), as it is described in section 3.6.9.

- Consequently, the authors considered superfluos (and potentially misguiding) a review of literature that goes too deep into the details of the CGE/I-O models.

- The new version of the paper includes an extended and more detailed model description that addresses this and some other remarks made by the referee #2. The new description includes workflow diagrams, a better characterization of the coping tactics, impacts calculations and cost distributions (section 3.2 and annex A).

- **Suggest you say who the paper is aimed at.**

Answer of the authors:

- The authors appreciate the suggestion of the referee #2. The paper is aimed at the whole community of researchers in economic impacts of natural catastrophes and economics of natural disasters.

- **Is this costs to agriculture as a share of total event cost?**

Answer of the authors:

- The interpretation of the referee #2 is correct.

- **Perhaps clarify that flood costs here include asset damage as well as income/expenditure impacts (an important aspects of agricultural flooding) Perhaps make it clearer that these two impact categories, farm production and off-farm commodity processing would potentially be treated as separate impact categories in flood assessment. This is said later but emphasize more here, I think.**

Answer of the authors:

- The new version of model description included in the paper makes more explicit the presence of commodity damage and expenditure/income impacts and their origins (section 3.2 and annex A).

- **Given actual cost data were collected it would be good to include absolute flood event cost estimates, and their make up/distribution between cost components.**

**A critical point is that that the quantitative results are given as a % of baseline: but what are the base line costs. The use of coefficients and weights to assess 'indirect' costs depend heavily on what the baseline estimate is > And assume that the baseline here is the sum of the two impact categories considered separately. I note that the estimates are by flood extent, but what are the costs per ha of vine flooded, or per unit capacity of wintery ?**

**Would be good to have some cost estimates, as suggested above, and this would help show the scale of the differences in the estimates with an without the connections**

Answer of the authors:

- The new version of the model description includes an extended description of impacts calculations in order to avoid confusions regarding how the COOPER model estimates impacts. As well, we have added a new section with the absolute cost estimates in the baseline scenarios.

- **Rather complicated to follow: rest on estimates of damage to assets plus impact on revenues and costs, including work in progress?**

Answer of the authors:

- The new version of model description included in the paper makes more explicit the presence of commodity damage and expenditure/income impacts and their origins. It also includes an extended description of the calculations of impacts (section 3.2 and annex A).

- **Seems to largely rest on the assumptions regarding the impact on the winery. Estimates of flooding on the wine production areas**

**can be based on ex 'farm gate' effects. The variation depends then mainly on the effects on the winery: so either the winery incurs 'direct' damage, because it is flooded or it indirect damage because, been though not flood, the quality or quality of supply is affected: so what are the impacts on these two elements in the supply/value chain? I think you are saying the underestimation is where the winery is safe from flooding, but takes a hit from not having grapes. But if it does flood, the impacts are assessed as a loss of contents and process. Hence why there is a big lift in your figures 4 and 4. You might make this (more) clear.**

Answer of the authors:

- The new description of the model COOPER addresses this the point in section 3.2.2 and annex A.3

- **Figure 2: what's the top dotted blue line.**

Answer of the authors:

- The accumulated number of plots at a given position.

- **The assumptions and configurations are challenging to follow, How representative are these configurations of what is observed in practice: is the size exposure configuration that gives the highest cost increase common ? or has the industry already adapted to flood risk?**

Answer of the authors:

- The information that could conduct us to be able to establish a representative configuration for the cooperative winemaking system are neither publicly available nor willingly divulged by the stakeholders. Thus we cannot establish a "standard" in relation to configurations and spatial locations.

- Furthermore, it is likely that "real life" configurations are very different from each other and sometimes overexposed small to mid size farms whereas other times big farms stocked riverine lands. That is why we think that the exercise done has interest: between the extremes, we have all potential "real life" situations. Nonetheless, the work we are presenting here is not an analysis of a particular system, during a particular flood. Rather, it is a plausible system based in "real world" systems facing a multitude of different floods, that shows us the misestimations in which we incur when we choose to ignore the links among entities in a local productive chain.

- These misestimations are all the more important when flood impacts assessments are utilized to design and calculate the mechanisms of monetary compensations to be implemented in rural communities that are used to protect urban and industrial areas from floods.

- The section 4 has been integrated in a new section "Method" (now, it is is section 3.3) and has been entirely restructured to make the section flow better.

- **It would be useful to produce a cost function that summarizes the type of costs, even better it would be good to produce estimates of costs showing the make up of the cost estimates for the different scenarios /configurations. There appears to be 'damage' to asset 'stocks' as well as to income/expenditure flows: what's the proportions of these. Not quite sure what is meant by material damage. Is there an underlying flood evet cost algorithm?.**

Answer of the authors:

- The new description of the model COOPER addresses this point (section 3.2.2 and annex A).

- **Concrete' flood, suggest rephrase.**

Answer of the authors:

- Rephrased as "given flood".

- **The results suggest, as far as I can see, that the main differences (either in costs by configuration or in costs relative to the baseline ) are due to autumn and winter flooding. What is the underlying seasonal distribution of flood costs ? More explanation of what to look for in the figures would be good, especially on observed variation (or lack of it)**

Answer of the authors:

- The new description of the model COOPER addresses this point (sections 3.2.1, 3.2.2 and annex A).

- **I think some of the points in the conclusions, might better go to reinforce the discussions: perhaps there should be a section on discussion of results and what they mean, in their and particularly, in practice, linked to the points made in the introductory sections.**

Answer of the authors:

- The new version of the article includes a section "Discussion" and a section "Conclusion"

- **It seems as though the cost estimates depend on seasonality as it determines where the grapes are in ex–vine storage and processing system, so the assessment of costs (relative to the baseline) largely depends on damage to stocks and flows of grapes in the system, which is seasonally defined. So I am asking why**

**would not a seasonally based estimate of damage accommodate this for the production (on the farm) and for the vinery, reflecting the dominant configuration. (A coping strategy might also be to important grapes from elsewhere to keep the process going, at a cost)**

Answer of the authors:

- The authors thank the referee #2 for the remark and the suggestion. As we understand the statement, we would like to point out that our work goes one step further of what the referee is proposing. Sure, we could make the assumption that stocks are either in one place or another -even using a probabilistic distribution behind based on a supposed dominant way of working- and do not take into account the flow of goods from one entity to another according to their schedules. This is somehow the exercise proposed in the case of *interactions within activities*: since information regarding losses of harvest does not flow from farm to winery, in those cases in which a flood impacts the winery, an estimation of the loss should be done and that might lead to problems of double accountability of damages.

- With the COOPER model we go one step further and we can know exactly where the stock is so estimations do not need to take place.

- Concerning the suggestion on the coping tactic, we agree that, in absolute terms, it would be an effective one. However the plausibility of such a tactic is inevitably linked to the reality of the CWS. As it is today, cooperative winemaking systems group farmers that decide to share the property of productive means to process their harvest and create their products employing professional experts for the day-to-day management and commercialization. The possibility that those farmers willingly accept to buy grapes from other producers is unlikely. Furthermore, if those farmers belong to a so-called *Appellation d'origine contrôlée*, they are subject to very strict rules regarding the quality and origin of the product (even the origin of the soil), if they want to keep the label.

- These institutional constraints made us decide to focus on the two more plausible coping tactics evoked by the interviewed farmers.

- **You say the approach is too costly: could estimates be built into the cost algorithm for representative configurations of the industry to allow for these so-called 'indirect' impacts.**

Answer of the authors:

- The idea we wanted to pass to the readers with such a statement is that the approach is too costly if it were to be implemented with the same amount of detail at a different level other than the local one, given the thoroughness with which the system is described and "translated" into a model. The best approach we can foresee is to analyze in depth different

local communities using detailed models, such as the COOPER, that feed regional or national models in some sort of nested structure.

- We have addressed this point in the conclusion of the new version of the paper.

H. C. Cochrane. Indirect losses from natural disasters: Measurement and myth. In Y. Okuyama and S. E. Chang, editors, Modeling Spatial and Economic Impacts of Disasters, Advances in Spatial Science, chapter 3, pages 37–52. Springer Berlin Heidelberg, 2004. doi:10.1007/978-3-540-24787-6_3.

C. Green, C. Viavattene, and P. Thompson. Guidance for assessing flood losses. CONHAZ WP6 Final Report, Flood Hazard Research Center - Middlesex University, Sept. 2011.

V. Meyer, N. Becker, V. Markantonis, R. Schwarze, J. C. J. M. van den Bergh, L. M. Bouwer, P. Bubeck, P. Ciavola, E. Genovese, C. Green, S. Hallegatte, H. Kreibich, Q. Lequeux, I. Logar, E. Papyrakis, C. Pfurtscheller, J. Poussin, V. Przyluski, A. H. Thieken, and C. Viavattene. Review article: Assessing the costs of natural hazards - state of the art and knowledge gaps. Natural Hazards and Earth System Sciences, 13(5):1351–1373, 2013.

E. C. Penning-Rowsell and C. H. Green. New insights into the appraisal of flood-alleviation benefits: (1) flood damage and flood loss information. Water and Environment Journal, 14(5):347–353, 2000. doi:10.1111/j.1747-6593.2000.tb00272.x.

**Summary of changes and comments regarding observations and recommendations from editor**

We thank the referee #1 for his feedback and suggestions for improving our manuscript. Below we provide our replies:

- **Data gathering: actually, I find the sources but not the type of data that you collected (We collected data from the Aude and Var administrative departments (southern France), both subject to major floods that have impacted the winegrowing sector . . . .). Could you please describe what kind of data are you talking about?**

Answer of the authors:

- Data sources and applications have been disclosed in a new table, included as "Annex B".

- **I agree with R2: the subsections should be reduced**

Answer of the authors:

- The whole article has been restructured to accommodate a more standard structure in scientific communications. Subsections have been reduced

- **A clear figure/diagram of the methodology is needed in order to understand the starting point and the way to reach the results. The research approach presented in the paper must be clear enough to be reproduced by other researchers.**

Answer of the authors:

- New section "Method" includes a workflow diagram of the method we propose.

- **Conclusions should be more concise and direct.**

Answer of the authors:

- The conclusions have been rewritten in the new version of the paper. Also a new section "Discussion" has been included.

---

## Referee Report (RR1)

[revised manuscript text omitted]

**Goal:** Do interactions matter in flood impact assessment?

**Method:** Comparative analysis of the current practice and simulations with explicit Interactions

**Baseline simulation:** Current practice

**Experiment I** Influence of explicit interactions between material entities

**Experiment parameter:**
3 levels of interaction
Fixed configuration of links (homogeneous)
Flood extent and season
Spatial coordinates of winery

**Experiment output:**
**IMP** for each type of interaction

**Experiment II** influence of heterogeneity in size and degree of exposure of farms

**Experiment parameters:**
6 configurations of links
Fixed level of interaction (full interaction)
Flood extent and season
Spatial coordinates of winery

**Experiment output:**
**IMP** for each type of configuration

Analysis of simulations

**Baseline simulation:** *no interaction* simulation from experiment I

Baseline scenario vs. rest of IMP from experiment I

Baseline scenario vs. IMP from experiment II

Workflow of impact simulator

input

Experiment parameters | Fixed spatial distribution of entities | Other parameters

simulator

COOPER model

Flood extent = 0?

yes — no

output

Business as usual scenario (BAU) | Flood scenario (FS)

impact

**IMP = FS - BAU**

Damage
- Soil
- Plant (crop)
- Building (equipment)
- Yield

Variation
- Wine-making cost
- Vine-growing cost

**Figure 2.** Workflow of the study

[Figure]

[revised manuscript text omitted]
 organized by source (rows) and element feature (columns). A check mark (✓) indicates that the source provides information on the corresponding feature.

| Category | Source | CW: Size | CW: Main Stages production | CW: Cost structure | CW: Financial data | CW: Damage function | CW: Sharing rules | CW: Behavior when impact | Farm: Size | Farm: Tasks | Farm: Cost structure | Farm: Financial data | Farm: Damage function | Farm: Behavior when impact | Plot: Productivity | Plot: Damage function | World: Number of plots | World: Number of farms | World: In/out flood prone area proportions | World: Market price |
|---|---|---|---|---|---|---|---|---|---|---|---|---|---|---|---|---|---|---|---|---|
| GIS | EEA (2012) | | | | | | | | | | | | | | | | | | ✓ | |
| GIS | IGN (2020) | | | | | | | | | | | | | | | | | | ✓ | |
| GIS | MTES (2020) | | | | | | | | | | | | | | | | | | ✓ | |
| Statistics | FranceAgriMer (2012) | ✓ | | | | | | | | | | | | | ✓ | | | | | ✓ |
| Statistics | Agreste (2010) | | | | | | | | ✓ | | | | | | | | | | | |
| Statistics | FADN (2014) | | | | | | | | | | ✓ | ✓ | | | | | | | | |
| Reports | INSEE (2016) | | | | | | | | | | | ✓ | | | | | | | | |
| Reports | CCMSA (2017) | | | | | | | | | | | ✓ | | | | | | | | |
| Reports | Centre d'économie rurale (2017) | | | ✓ | | | | | | | | | | | | | | | | |
| Reports | Centre d'économie rurale (2014) | | | | | | | | | | ✓ | | | | ✓ | | | | | |
| Literature | Chevet (2004) | | ✓ | | | ✓ | | | | | | | | | | | | | | |
| Literature | Folwell and Castaldi (2004) | | | ✓ | | | | | | | | | | | | | | | | |
| Literature | Battagliani et al. (2009) | | | | | | | | ✓ | | | | | | | | | | | |
| Literature | Brémond (2011) | | | | | | | | | ✓ | | | | | | | | | | |
| Literature | Biarnès and Touzard (2003) | | | | | | ✓ | | | | | | | | | | | | | |
| Literature | Jarrige and Touzard (2001) | | | | | | ✓ | | | | | | | | | | | | | |
| Literature | Touzard et al. (2001) | | | | | | ✓ | | | | | | | | | | | | | |
| Interviews | Experts | | | | ✓ | ✓ | | ✓ | ✓ | ✓ | | | | | | ✓ | | | | |
| Interviews | Winery CEOs | ✓ | ✓ | | | | ✓ | ✓ | | | | | | | | | ✓ | ✓ | | |
| Interviews | Vinegrowers | | ✓ | | | | | | | | | | | | ✓ | | | | | |

CW = Cooperative winery

**Table B1.** Sources of information organized by element feature

---

## Author Response (AR2)

**Summary of changes and comments regarding observations and recommendations from referee #2**

We thank the referee #2 for his new feedback and insights to improve the quality of our manuscript. Below we provide our replies to both the general comments and to the specific points made by the referee, using the following structure: the referee's comment is highlighted in bold font, whereas the answer of the authors is included in normal font.

**Referee's general comments**

- **Title: suggest adding ':the case of winemaking in France'**

  *Answer of the authors:*

  – The recommendation has been taken into account and the title modified.

- **Costs definitions and classifications:I go back to my original point that in my view should be addressed in the manuscript. As the commentary in the introduction implies, there is often lack of clarity in the definition, classification and treatment of costs, and this can lead to a combination of confusion, uncertainty and misinterpretation, The topic of the paper is central to this, dealing as it does with 'indirect flood costs'. Definitions of flood costs and treatment of indirect costs have varied. In this context, the authors need to clarify and explain the typology/classification they are using, and how the classification and estimates of costs fit (or does not) with the conventions used by others. There is an important methodological link here. Furthermore, and important in my view, the authors should return to this point in the discussion showing how their approach and estimates relate to the review of the literature on this topic at the beginning of the paper. A critical point also is whether taking the definition of indirect costs as they do makes a 'significant' difference to the estimate of flood damage costs, and in what circumstances. The authors refer to this later, but again this could be framed in terms of this opening commentary.**

  **Detail of cost definitions and classifications, including make up of costs. Detail is now provided in a long appendix, to which limited reference is actually made. There needs to be more explanation in the manuscript itself, with more information summarised in the table on 'indicators', please see comments. Is the appendix really supplementary information.**

  *Answer of the authors:*

  – The authors have made different modifications in the manuscript to accomodate the referee #2's requirements.
  – Section 3.1 now informs the reader of a classification adopted to present a decomposition of the baseline results into direct and indirect impacts.
  – In section 4.1, the baseline results are analyzed according to this new classification. We use the resulting decomposition to show the reader the gap between direct and total damages.
  – In sections 4.2.2 and 4.3.2 the quantitative analyses now include absolute values, that, according to our classification, represent variations in the estimation of the indirect damages.
  – More references are made to the appendix throughout the text.

- **The point should be made, in the manuscript and in the abstract, that the modelling and the results are synthetic and not based on actual recorded or observed costs. (it would be nice to anchor them to observations on actual costs, from other work).**

  *Answer of the authors:*

  – A new statement disclosing that flood impacts are not based in real cases (namely we are not modeling a specific event) has also been included in section 3.2.1 (*Flood process: intensity and impacts of floods*)
  – The authors have not modified the abstract. They consider that to refer to their method as a *virtual laboratory for the ex ante estimation of impacts* already indicates readers that results are based on simulations.

- **Presentation of results. I remain of the view that estimates of absolute costs for the scenarios, summarised in a table, would be helpful, together with a profile that shows the distribution of (synthetically generated) costs by type. %s have obvious limits. A table of results would better support the points made in the discussion. The key question: whether this type of assessment makes a difference and under what circumstances could then be more clearly answered.**

  *Answer of the authors:*

– A new table with total and direct impacts has been included in section 4.1. This table includes the detail of damages by season in absolute and relative terms, as well as a classification depending on whether the winery is flooded or not.

**Referee's specific points on the text**

- **Lines 27-41: I go back to my original point, that in my view should be addressed in the manuscript : as the commentary here implies there is often lack of clarity in the definition, classification and treatment of costs, and this can lead to a combination of confusion, uncertainty and misinterpretation, The topic of the paper is central to this, dealing as it does with 'indirect flood costs', but definitions of this and treatment of these has varied. In this context, the authors need to clarify and rationalise the typology/classification they are using, and how the classification and estimates of costs fits (or does not) with the conventions used by others. There is an important methodological link here. Furthermore, and important in my view, the authors should return to this point in the discussion showing how their approach and estimates relate to the review of the literature on this topic at the beginning of the paper. A critical point also is whether taking the definition of indirect costs as they do makes a 'significant' difference to the estimate of flood damage costs, and in what circumstances. The authors refer to this later, but again this could be framed in terms of this opening commentary.**

  *Answer of the authors:*

    – See answer of the authors to referee #2's second general comment.

- **Line 39: This sentence should begin with 'However'.**

  *Answer of the authors:*

    – The sentence has been modified

- **Line 167: I missed this at first because when I searched there is no annex B : it is an Appendix. I think the links to this appendix can be strengthened throughout this, including in the section on the costs function and the table of indicators.**

  *Answer of the authors:*

    – The sentence has been corrected.

- **Line 171 (in reference to figure 2): This is not the workflow of the paper, as such, but of the modelling approach.**

  **Figure 2's caption: A more detailed title is required**

  *Answer of the authors:*

    – Phrase restated: "The approach we follow in this paper is outlined in figure..."
    – The figure 2's caption has been modified

- **Line 175: Now we have a more detailed description, which is good, it would be good to briefly say how this model compares with similar models of its type in feature or application, and importantly why it was considered to be appropriate for the purpose, and the limitations of the model (linked to the limits of the study dealt with later)**

  *Answer of the authors:*

    – An Overview of agent-based models applied to the study of floods has been included in the introduction.
    – The section '*overview*' now includes a highlight of the reasons to choose ABMs as simulation method.
    – We agree that a comparison of models would be interesting. However as it has now been highlighted in the introduction, at the time this work was done, there were no similar models to the COOPER model. That was one of the reasons to build the COOPER model from scracht
    – As for the limits, we consider that they have already been disclosed in the discussion section, constituting new research lines.

- **Figure 3's caption: A more detailed title is required**

  *Answer of the authors:*

– The figure 3's title has been modified

- **Line 224: you mean 'Output' not yield, which is a ratio I now see at the end of the paper, the appendix uses the term yield, where yield presumably is taken to mean Output/Farm, a ratio, perhaps make it clear we are talkng about yield at the farm scale.**

  **Line 225: Ditto and below.**

  **Line 271: output or production loss.**

  *Answer of the authors:*

  – We agree with the referee #2 that it is far more common to use the term 'output' rather than the term 'yield' in an economic paper. However we made the deliberate choice of keeping the term 'yield' to easily differentiate between the output from the plot/s from the output of the winery.
  – The 'yield' of a given farm in the context of the COOPER model is not a ratio but the sum of the "yield or output" harvested in each plot owned by said farm. In other words, as it happens in real life, in the COOPER model each plot "grows" grapes during spring and summer that are harvested in autumn. The sum of the amount harvested from each plot owned by a given farm is our fam's yield (qi). Then all fams give their yield to the winery, that produces the wine (which is the system's production -Q) and sells it. Finally the revenue from the wine selling is distributed among farms according to the ratio qi/Q.

- **Line 274: this takes us back to the definition and classification of costs, asset damage, revenue losses, extra operating costs l the term damage implies 'asset/capital' loss, whereas many many of these costs are 'revenue' items.**

  *Answer of the authors:*

  – Section 3.1 now informs the reader of a classification of damages adopted for the paper. In section 3.2.2 we use table 1 to explicitly state which impacts are considered direct and which impacts are indirect according to the classification of section 3.1. As well, section 4.1 offers a table with absolute and relative values of total and direct damages for the baseline scenario.

- **Line 359 (referred to section 3.2.10): is this really flood cost estimation?. How does the classification of costs align with the discussion at the beginning, how and why are costs identified and grouped in this way? what are the main constituents of these cost element that make up the function, and are they independent? There is no reference to the Appendix here that contains the details. The appendix seems to be more supplementary data than an appendix as such. There could be a succinct summary of the cost components and their constituents. The editor may have a view, but I think there should be more detail in the main manuscript on the costs, asset damage, changes in annual revenues and expenses and so on, and some comment on what are regarded as direct and what are indirect in the classification commonly used. This could be done by strengthening the indicator table with more detail on the cost element, see below.**

  *Answer of the authors:*

  – The authors use a simulation model that allow them to simulate productive processes with and without presence of floods. By comparing the simulations done without floods with the simulations done with floods, the authors consider that the model captures the whole cost of a given flood within the boundaries of the system.
  – The main constituents of these impacts are detailed in Table 1. so, for instance, impacts on investments are constitued by the reposition cost of damages in buildings, cost of soil reconditioning and the difference between the planned replanting investment in the scenarios without flood and the replanting investment forced by the flood.
  – The ten indicators have been choosen prior to the construction of the COOPER model as relevant indicators regarding impacts of floods in agricultural supply chains and, more concretely, in a french cooperative winemaking system. They are all considered independent in the sense that they are not calculated from one another. Notwithstanding, insofar as we are in production chain losses of yield are going to provoke variations in production costs and so on. Thus there exist some sort of domino effect linking them.
  – Table 1 now includes information on the metric for each indicator as well as the classification in direct or indirect impact.

- **Line 362: there needs to be a reference here to the Appendix that provides further explanation : I didnt see this until i reached the end.**

  *Answer of the authors:*

  – The text includes now more references to the appendix

- **Table 1 (page 16, line 393): the title is unhelpful. Need a full title that explains contents.**

  **Table 1 (page 16, line 393): More detail would help here, and some further explanation. I now see there are detials in the appendix, which is not signposted here. I think that more information on the metrics of these cost items and sources of data should be summarised here in this table for the main part of the manuscript. Damages implies asset losses and some of these impact are, but some are losses in revenue. it would be useful to inlcude a colums that shows the units of measurement and metrics for each of these high level indicators, eg soil damage assessed in terms of soil restoration costs, and for the other items also, them it becomes clear what is being measured. Also the table could show where the data are derived from**

  *Answer of the authors:*

  – The title of the table has been modified.
  – A new column (named *"Metric"*) has been included in the table. Inasmuch as all indicators eventually provide the information in monetary terms, a mention of the measurement unit has been included in the table's caption.
  – A new column specifying whether the indicator is considered direct or indirect impact has been included.

- **Line 407: advise avoid use of the term 'concrete' in English unless the authors mean a mixture of sand, stone and cement, which they may for buildings perhaps, but not here. Also see below**

  **Line 427: as above, implies a strong mix. . . .**

  *Answer of the authors:*

  – The phrases have been restated avoiding the term.

- **Line 544: (referred to section 4.1): the authors are missing a trick here. There is much interest in assessing the profile and make up of costs, and the balance of asset and other costs, and how the estimated costs align with the kind of classifications referred to earlier, by other authors and in this work. A table on the type and distribution of costs would be really useful. This would also show where the big hits are, and from a resilience/response aspect, where the effort should be placed in terms of Flood risk management.**

  **Line 544: (referred to section 4.1): which is what, max €8 million in figure 6 ?**

  *Answer of the authors:*

  – The article now includes a fully rewritten section 4.1 where we enter in the detail of the magnitude of total impacts as well as direct ones. A new table with total and direct impacts has been included in section 4.1. This table includes the detail of damages by season in absolute and relative terms, as well as a classification depending on whether the winery is flooded or not.

- **Line 646: is this apparent in the Quant results - if they are presented, perhaps for the extreme cases.**

  **Line 653: see my earlier comment, where are the quantitative results that show the breakdown and distribution of costs, and which of these these costs significant in absolute terms. I am still trying to understand whether allowing for the linkages makes a difference and is worth the trouble, when the winery itself is not flooded but farms are, and vice versa. Also, some of the literature alluded to use 'weights' applied to direct costs, to estimate knock on effects in the non flooded areas ; are there implicit weights here? I am trying to see where the comment below is coming from in the results, namely: 'the observation does not allow us to give a general recommendation one way or the other . . . .'**

  *Answer of the authors:*

- The authors have provided orders of magnitude in absolute values for all the section of quantitative analysis. We expect that this new information helps readers to realize the significativity of the differences especially when compared to the annual potential added value of the system.

- **Line 779: this begs the question what decisions might be made if these assessment/information gaps were filled ; are these estimation errors 'significant' in the scheme of things, relatively or absolutely This would help decide whether further assessment effort is justified would changes in future risks, possibly associated Climate Change, make the extra effort more or less valid.**

  *Answer of the authors:*

  - The authors consider that the new elements added (explicit impact classification, table with absolute total and direct values of damages for the baseline and explicit numerical values in both quantitaive analyses) reinforce the opening argument of the conclusion: *"[. . . ] the introduction of explicit interactions in productive systems has a non-negligible impact on the amount of damage estimated at a microeconomic scale".* Indeed, the inclusion of absolute and relative values for both total and direct damages now gives a clearer idea of the magnitude of the damages that might be incorrectly estimated. Furthermore, according to the classification established in our work, those incorrectly estimated damages are going to be indirect damages and their magnitude is going to depend on different factors such as, e.g., season, flood extent, material entities flooded, links among material entities, etc.

- **Line 800: or other responses such as relocation.**

  *Answer of the authors:*

  - Indeed. Different responses to cope or to adapt in order to avoid bankrruptcies are to be expected.